# Two-billion-year-old volcanism on the Moon from Chang'e-5 basalts

Qiu-Li Li[1,6], Qin Zhou[2,6], Yu Liu[1], Zhiyong Xiao[3], Yangting Lin[4], Jin-Hua Li[4], Hong-Xia Ma[1], Guo-Qiang Tang[1], Shun Guo[1], Xu Tang[4], Jiang-Yan Yuan[1], Jiao Li[1], Fu-Yuan Wu[1], Ziyuan Ouyang[5], Chunlai Li[2✉] & Xian-Hua Li[1✉]

The Moon has a magmatic and thermal history that is distinct from that of the terrestrial planets[1]. Radioisotope dating of lunar samples suggests that most lunar basaltic magmatism ceased by around 2.9–2.8 billion years ago (Ga)[2,3], although younger basalts between 3 Ga and 1 Ga have been suggested by crater-counting chronology, which has large uncertainties owing to the lack of returned samples for calibration[4,5]. Here we report a precise lead–lead age of 2,030 ± 4 million years ago for basalt clasts returned by the Chang'e-5 mission, and a $^{238}$U/$^{204}$Pb ratio ($\mu$ value)[6] of about 680 for a source that evolved through two stages of differentiation. This is the youngest crystallization age reported so far for lunar basalts by radiometric dating, extending the duration of lunar volcanism by approximately 800–900 million years. The $\mu$ value of the Chang'e-5 basalt mantle source is within the range of low-titanium and high-titanium basalts from Apollo sites ($\mu$ value of about 300–1,000), but notably lower than those of potassium, rare-earth elements and phosphorus (KREEP) and high-aluminium basalts[7] ($\mu$ value of about 2,600–3,700), indicating that the Chang'e-5 basalts were produced by melting of a KREEP-poor source. This age provides a pivotal calibration point for crater-counting chronology in the inner Solar System and provides insight on the volcanic and thermal history of the Moon.

Even though mare basalt covers only roughly 17% of the surface of the Moon[8], its protracted formation record spans more of the lunar magmatic history than any other geological unit. Radioisotope age studies of basaltic samples returned by the Apollo and Luna missions and lunar meteorites have revealed that basaltic magmatism on the Moon occurred between around 4.4 billion years ago (Ga) (ref. [9]) and around 2.9–2.8 Ga (refs. [2,3]), with two major pulses around 3.95–3.58 Ga and 3.38–3.08 Ga (ref. [10]). However, crater-counting chronology suggests a more extended period of basalt volcanism occurring between around 4.0 Ga and around 1.2 Ga (refs. [4,11]). Some of the youngest mare basalt units in Oceanus Procellarum have been estimated to be around 2.2–1.2 Ga (refs. [4,11,12]), which would putatively expand the duration of mare volcanism to over roughly three billion years. Whether or not mare volcanism continued to around 2.2–1.2 Ga, or perhaps even younger in small eruptions[5], has long been a major question. These young crater-counting ages, however, have not been confirmed by radioisotopic ages. The actual end of mare volcanism has not yet been constrained by radioisotopic dating owing to the lack of available samples from these younger volcanic units, which have incurred fewer impacts.

## Age of Chang'e-5 basalt

Precise and accurate age determination of these young mare basalts is crucial not only for unravelling the timing and duration of lunar

volcanism but also for investigating late-stage basaltic petrogenesis and the melting of lunar mantle sources that are relevant to the thermal and chemical evolution of the Moon. The landing site of Chang'e-5, China's first lunar-sample-return mission, was selected because it is located on one of the youngest mare basalt units northeast of Mons Rümker in northern Oceanus Procellarum[13]. Therefore, basalts returned by Chang'e-5 provide an opportunity to understand the timing and mechanism of one of the youngest units of mare volcanism on the Moon. In addition, precise radioisotopic dating of these newly returned basalts has a critical potential to verify and calibrate lunar impact crater-counting chronology, which is the main basis for dating most geological units of the other inner Solar System bodies.

The Chang'e-5 samples studied in this research were scooped from the lunar regolith surface and include three one-inch epoxy mounts (samples CE5C0000YJYX041GP and CE5C0000YJYX042GP with two basalt clasts larger than 1.5 mm in each, and sample CE5C0800YJFM00102GP with 20 mg of soil) and two aliquots of soils (samples CE5C0100YJFM00103 of 1 g and CE5C0400YJFM00406 of 2 g) allocated by the China National Space Administration. Around 800 lithic clasts (greater than 0.25 mm) were randomly picked from the two soil samples to make additional epoxy mounts. The lithic clasts in the soils are composed of about 45% basalt and about 55% breccia. The breccia clasts are dominated by basalt fragments (more than 80%) with minor impact melt and agglutinate. The majority of the basalt clasts

[1]State Key Laboratory of Lithospheric Evolution, Institute of Geology and Geophysics, Chinese Academy of Sciences, Beijing, China. [2]Key Laboratory of Lunar and Deep Space Exploration, National Astronomical Observatories, Chinese Academy of Sciences, Beijing, China. [3]Planetary Environmental and Astrobiological Research Laboratory, School of Atmospheric Sciences, Sun Yat-sen University, Zhuhai, China. [4]Key Laboratory of Earth and Planetary Physics, Institute of Geology and Geophysics, Chinese Academy of Sciences, Beijing, China. [5]Center for Lunar and Planetary Sciences, Institute of Geochemistry, Chinese Academy of Sciences, Guiyang, China. [6]These authors contributed equally: Qiu-Li Li, Qin Zhou. ✉e-mail: licl@nao.cas.cn; lixh@gig.ac.cn

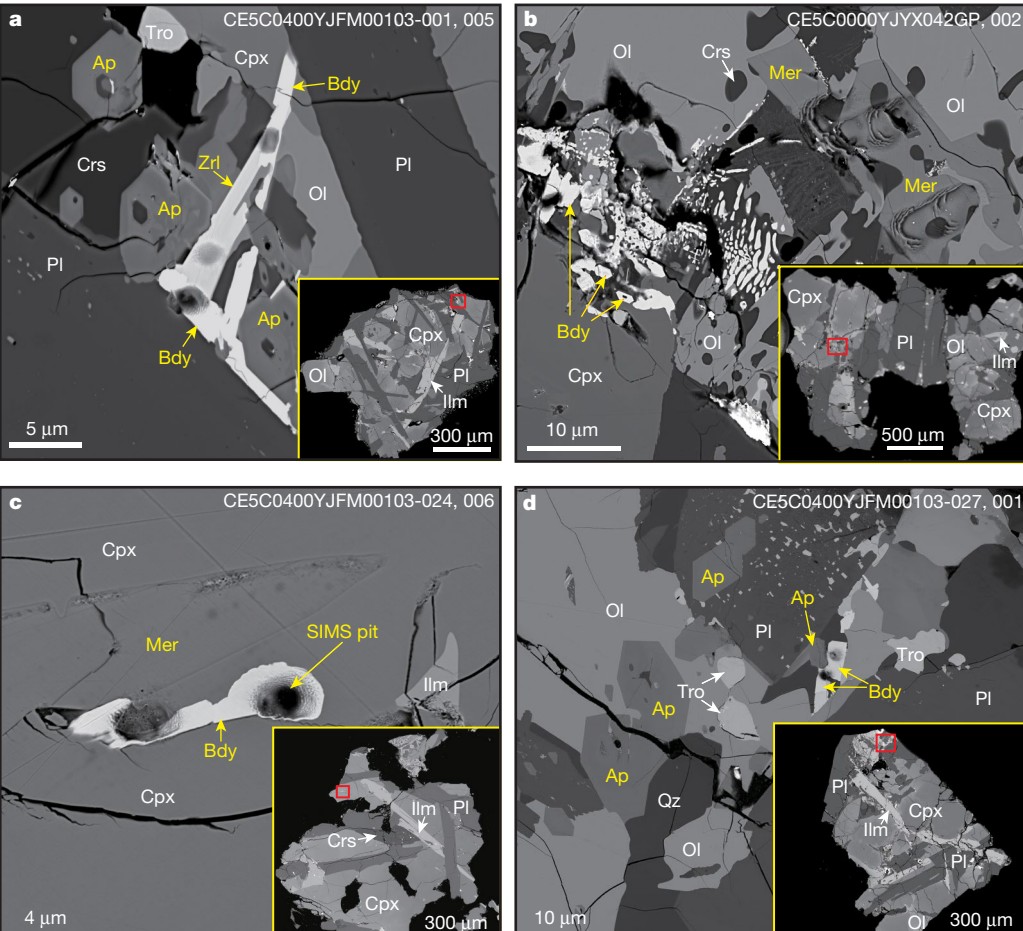

**Fig. 1 | Back-scattered electron images of representative dated minerals in the Chang'e-5 basalt clasts. a**, Intergrown texture of baddeleyite, zirconolite and apatite in a subophitic basalt clast. **b**, Subhedral baddeleyite and merrillite occur within mesostasis composed of Fe-rich olivine, clinopyroxene, K-feldspar and troilite in a poikilitic clast. **c**, Baddeleyite and merrillite in a subophitic clast. **d**, Euhedral apatite and baddeleyite occur as intergranular phases associated with Fe-rich olivine in an equigranular basalt clast. Pits in Zr-bearing minerals are the in situ analytical spots from SIMS. The areas of dated minerals in the clasts are outlined (red rectangles) in the corresponding insets. Ap, apatite; Bdy, baddeleyite; Cpx, clinopyroxene; Crs, cristobalite; Ilm, ilmenite; Mer, merrillite; Ol, olivine; Pl, plagioclase; Qz, quartz; Tro, troilite; Zrl, zirconolite. Codes in the top right corner indicate sample name, mount number and grain number on the mount (**a**, **c**, **d**), or mount name and grain number (**b**, note: this is a one-inch epoxy mount sample with two basalt grains on it.).

(about 80%) show subophitic and poikilitic textures, with the remainder being porphyritic and equigranular textures. The basalt clasts have various grain sizes (approximately 10–600 μm), but similar major mineral constituents of clinopyroxene, plagioclase, olivine, ilmenite, and rare troilite and cristobalite (Fig. 1). The euhedral-to-subhedral phosphate minerals apatite and merrillite are found in all types of basalt clast (Fig. 1, Extended Data Figs. 1, 2, Supplementary Table 1) and commonly occur along the margins of iron (Fe)-rich olivine, clinopyroxene and ilmenite. The minor zirconium (Zr)-bearing minerals baddeleyite, zirconolite and tranquillityite are fine-grained (3–8 μm), euhedral to subhedral in shape, interstitial-to-major silicate phases, and common in coarse-grained (greater than 100 μm) poikilitic, equigranular and subophitic basalt clasts, but not in porphyritic basalt clasts (Fig. 1, Extended Data Figs. 1, 2, Supplementary Table 2). In some cases, Zr-bearing and phosphate minerals show intergrowth textures with each other (Fig. 1a, c, d, Extended Data Fig. 2), which suggests that they formed during the same final crystallization stage of the magma. Forty-seven representative basalt clasts with various textures were used for radioisotopic dating.

The uranium (U)–lead (Pb) isotopic compositions of various mineral phases in Chang'e-5 basalt clasts were determined using a CAMECA IMS 1280HR secondary ion mass spectrometer (SIMS) (complete dataset presented in Supplementary Table 3). Zr-bearing minerals, phosphates and others (that is, plagioclase, pyroxene and matrix minerals) were analysed using a primary O⁻ beam with roughly 3-μm, roughly 8-μm and roughly 30-μm spot sizes, respectively (Methods). Pb isotope analyses on 17 poikilitic clasts, 18 subophitic clasts and 10 equigranular clasts (Extended Data Table 1) were used to construct three Pb–Pb leftmost isochrons[10]. The radiogenic $^{207}Pb/^{206}Pb$ ratios of $y$ intercepts are translated to Pb–Pb ages of 2,027 ± 7 million years ago (Ma) (95% confidence level, and hereafter except where otherwise noted), 2,030 ± 6 Ma and 2,034 ± 8 Ma, respectively (Extended Data Fig. 3a–f). Forty-four Pb isotope analyses were conducted on plagioclase, pyroxene and matrix from two fine-grained porphyritic clasts (without visible Zr-bearing minerals) and yielded a leftmost isochron Pb–Pb age of 2,027 ± 54 Ma (Extended Data Fig. 3g, h). Despite the distinct petrographic textures in Chang'e-5 basalts, these four isochrons have consistent $y$ intercepts and slopes within uncertainties (Extended Data Fig. 3), indicating both their identical age and derivation from most probably the same source. Taken together, a total of 159 analyses with negligible terrestrial Pb contamination for various mineral phases form an integrated isochron yielding a Pb–Pb age of 2,030 ± 4 Ma (Fig. 2). The age is interpreted as the best estimate of the crystallization age of the Chang'e-5 basalts given that all the clasts studied here show pristine magmatic textures without evident overprinting from shock metamorphism (Fig. 1). This age represents the youngest crystallization age reported so far

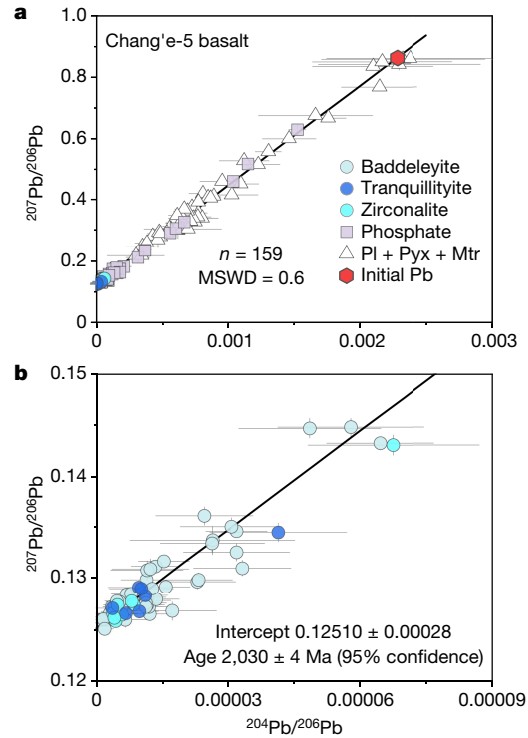

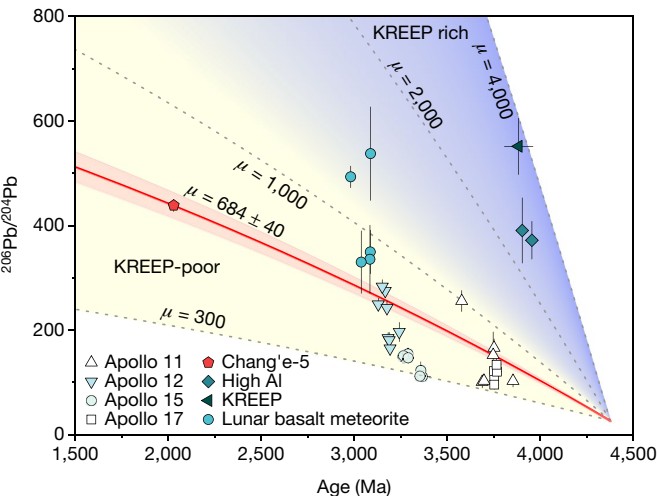

**Fig. 3 | Initial $^{206}Pb/^{204}Pb$ ratios versus crystallization ages of the lunar basalts and meteorites.** The lines represent the two-stage Pb isotope evolution of lunar mantle sources at given $\mu$ values[6]. The gradient areas are associated with KREEP-poor (yellow) to KREEP-rich (blue) mantle sources according to $\mu$ values. The Apollo and meteorite data are from refs. [6,7,10]. Error bars are $2\sigma$ s.e. A11, Apollo 11 high-Ti basalts; A12, Apollo 12 low-Ti basalts; A15, Apollo low-Ti basalts; A17, Apollo 17 high-Ti basalts; High-Al, Apollo 14 high-Al basalts; KREEP, Apollo 15 KREEP basalts; Lunar basalt meteorite, low-Ti and very-low-Ti basaltic meteorites (NWA 4734 and NWA 773 clan).

**Fig. 2 | Pb–Pb isochron for the Chang'e-5 basalts. a**, The integrated Pb–Pb isochron showing the mixing line between the $y$ intercept as radiogenic $^{207}Pb/^{206}Pb$ and the initial Pb compositions ($^{204}Pb/^{206}Pb = 0.00228 \pm 0.00011$, $^{207}Pb/^{206}Pb = 0.860 \pm 0.019$). **b**, The enlarged lowest part of the isochron in **a** highlighting the measurements of Zr-bearing minerals. The black line is the best-fitted isochron with an equation of $y = (323 \pm 7)x + (0.12510 \pm 0.00028)$. Error bars represent $1\sigma$ s.e. Mtr, matrix; Pl, plagioclase; Pyx, pyroxene.

for lunar basaltic rocks by the radiometric method and thus extends the range of radioisotopic ages of lunar basalt by about 800–900 Myr. Therefore, this study provides conclusive evidence that magmatic activity on the Moon persisted until at least 2 Ga. This insight into the existence of this youngest-known volcanism provides a critical constraint for understanding the thermal mechanisms behind the longevity of lunar magmatism.

## Mantle source signature

One of the leading mechanisms considered for sustaining such young lunar volcanism is potassium, rare-earth elements and phosphorus (KREEP)-related radiogenic heating in the basalt source[1,3,14]; thus characterizing the geochemistry of the mantle source of this youngest-dated mare basalt reported so far provides a critical test of this hypothesis. The initial Pb isotopic composition of the basalts and corresponding time-integrated $\mu$ value ($^{238}U/^{204}Pb$ ratio) for their mantle source can fingerprint the Chang'e-5 basalt mantle source and its chemical evolution[6,7,10]. Five of the 106 U/Pb analyses for rock-forming minerals (mainly plagioclase) show negligible U (with $^{238}UO^+/^{208}Pb^+$ ratios of <0.01) and clustered $^{207}Pb/^{206}Pb$ values of 0.855–0.872 (Supplementary Table 3). Thus, their weighted mean of $0.860 \pm 0.019$ (2 s.e., $n = 5$, mean squared weighted deviation = 0.11) provides the best estimate for the initial $^{207}Pb/^{206}Pb$ ratio of the Chang'e-5 basalts (Extended Data Fig. 4). The initial $^{204}Pb/^{206}Pb$ ratio, however, is difficult to measure precisely and accurately owing to extremely low $^{204}Pb$ counts (less than 0.05 counts per second (cps)) (Supplementary Table 3). Alternatively, it can be calculated as $0.00228 \pm 0.00011$ (2 s.e.) based on the best-fit Pb–Pb isochron (Fig. 2a) and the aforementioned best estimate of the initial $^{207}Pb/^{206}Pb$ ratio. This calculated initial $^{204}Pb/^{206}Pb$ ratio is consistent

within errors with the weighted mean of $0.00235 \pm 0.00043$ (2 s.e.) for the two lowest measured $^{204}Pb/^{206}Pb$ ratios (Supplementary Table 3), justifying the rationale of the calculation.

Determination of the $\mu$ value of a basalt mantle source is dependent on the lunar Pb-isotope evolution model[6,7,10]. On the basis of the lunar magma ocean (LMO) model[15] that presumably generated all major lunar silicate reservoirs, including the sources for lunar basalts, a two-stage model for lunar Pb-isotopic evolution is proposed[6,7]. The evolution of the Pb isotopes of a basalt source starts from $t_0$ (around 4,500 Ma)[16] for Moon formation with $\mu_1 = 462 \pm 46$ for the LMO[6], to $t_1$ (around 4,420–4,300 Ma)[17,18] for LMO crystallization and the formation of major geochemically distinct reservoirs with different $\mu_2$ values, to $t_2$ for mare basalt formation with the initial Pb isotopes. Although $t_1$ remains uncertain, its age range has little effect on the calculated $\mu_2$ values (Extended Data Fig. 5). Thus, for comparison with previous results[6,7], a $t_1$ of $4,376 \pm 18$ Ma was selected to calculate a two-stage $\mu$ value of $684 \pm 40$ for the Chang'e-5 basalt source (Fig. 3). This $\mu$ value is within the range ($\mu \approx 300–1,000$) of the low-titanium (Ti) and high-Ti Apollo basalts, but considerably lower than those ($\mu \approx 2,600–3,700$) of the KREEP and high-aluminium (Al) basalts[6,7] (Fig. 3). This marked difference suggests that the Chang'e-5 basalts from the Procellarum KREEP Terrane were most probably produced by the melting of a KREEP-poor source. An apparent increase in $\mu$ values from around 3.4–3.0 Ga for low-Ti Apollo basalts and low-Ti and very-low-Ti basaltic meteorites (NWA 4734 and NWA 773 clan) suggests a progressive contribution of a KREEP-like component in such rocks[6,7,10]. However, the Chang'e-5 basalts do not follow this trend, indicating that KREEP-like components were not involved in our samples, neither in the deep source nor during shallow contamination of the KREEP material. Corroborating evidence for a non-KREEP source for the Chang'e-5 basalts is provided by strontium–neodymium (Sr–Nd) isotopes[19], but the results in this study using the radioactive element U offer direct evidence for heat-producing elements not being concentrated in the Chang'e-5 basalt mantle source. Thus, these results strongly suggest that the idea of KREEP-induced heating[1,14,20] for the generation of these young lunar magmas requires further investigation or the consideration of other mechanisms.

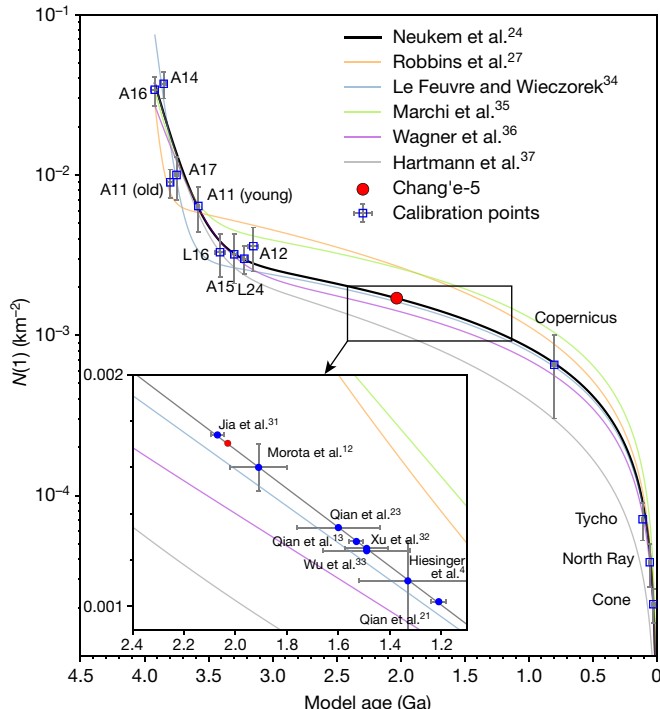

**Fig. 4 | The critical reference point of the radioisotope age of the Chang'e-5 basalts for the lunar crater-counting chronology.** The lines are colour-coded according to different models of the crater-counting chronology of the Moon. The red dot marks the radioisotope age of the Chang'e-5 basalts and the translated crater density based on the crater-counting chronology function of ref. [24]. The blue squares are the calibration points established from Apollo and Luna samples[27–30]. The inset shows the various crater densities and model ages predicted for the mare unit at the Chang'e-5 landing site (blue dots)[4,12,13,21,23,31–33]. Error bars are 1σ s.e. N(1), number of craters with diameter > 1 km. Different models in the main panel are from refs. [24,27,34–37].

## Anchor point for cratering chronology

Radioisotope ages provide the yardstick for calibrating the age information obtained from crater statistics. The mare unit on which Chang'e-5 landed features both homogeneous reflectance spectra and surface morphology as seen from orbit[21], and ballistic sedimentation modelling[22,23] suggests a dominance of local mare basalts in the surface regolith. The basalt clasts returned show uniform geochemical characteristics[19] and a consistent radioisotope age of 2,030 ± 4 Ma (Fig. 2), convincingly pointing to an affinity with the mare unit of the landing site. Therefore, our radioisotope age obtained for the newly returned Chang'e-5 samples offer an opportunity to confirm the first-order reliability of the lunar crater-counting chronology established by the Apollo and Luna missions[24]. The calculated model ages for the mare unit on which Chang'e-5 landed mostly range between around 2.2 Ga and 1.5 Ga using the prevailing crater-counting chronology of ref. [24], within a difference of about 20% compared with the measured radioisotope age (Fig. 4). This difference is surprisingly consistent with that derived from a comparison of the current impact flux on the Moon from observations[25] and prediction by crater-counting chronology[24].

Substantial differences nonetheless exist among reported crater densities for the mare unit on which Chang'e-5 landed (Fig. 4), albeit a relatively simple geological context and the same crater-counting chronology model were used. Therefore, there is much potential to improve the accuracy of predictions by crater statistics. For the existing crater-counting chronology, Apollo and Luna samples have provided an initial database for ages ranging from around 4.0 Ga to 3.1 Ga as well as those younger than around 1 Ga (Fig. 4). The age of 2.03 Ga obtained

for the Chang'e-5 basalts resides squarely in the centre of this large gap (Fig. 4), fulfilling the long-sought-after goal to bridge the unanchored middle portion of the lunar crater-counting chronology[24] and improving this critical tool for dating unsampled surfaces on the Moon[26], as well as for translating the lunar crater-counting chronology to the other planetary bodies[24].

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

# Methods

The sample mounting, scanning electron microscope (SEM), electron probe microanalysis (EPMA) and SIMS analyses were performed at the Institute of Geology and Geophysics, Chinese Academy of Sciences (IGGCAS) in Beijing, China.

## SEM analysis

The Chang'e-5 basalt clasts studied were first embedded in epoxy mounts and then polished using a grinder. A Thermo Scientific Apreo SEM equipped with an energy dispersive spectroscopy (EDS) detector was used to identify the Zr-bearing and phosphate minerals. High-resolution back-scatter electron (BSE) imaging and semi-quantitative EDS analyses were conducted using a Zeiss Gemini 450 field-emission environmental SEM. For a large-scale BSE imaging of a single basalt clast, the measurement was performed at an acceleration voltage of 15 kV and a current of 2.0 nA, with a working distance of about 8 mm. For dated minerals in a localized area, the analyses were operated with an acceleration voltage of 5 kV, a beam current of 1.0 nA and a working distance of about 7 mm. The phosphate (apatite and merrillite) and Zr-bearing minerals (baddeleyite, zirconolite and tranquillityite) were examined by EDS (Extended Data Fig. 1).

## EPMA analyses for dated minerals

Both Zr-bearing minerals and phosphate were analysed using a CAMECA SXFive FE electron probe microanalyser. An acceleration voltage of 20 kV and beam current of 30 nA were used for all analyses, with a spot size of 1 µm. Data were processed with the phi–rho–Z matrix correction using CAMECA PeakSight software (version 6.2). Synthetic glasses (single REE oxide–calcium oxide–aluminium oxide–silicon dioxide) from P&H Developments were used as the standards for REE. The standards used for the other elements analysed were periclase (magnesium (Mg)), K-feldspar (aluminium (Al)), rhodonite (silicon (Si), calcium (Ca) and manganese (Mn)), rutile (Ti), chromium oxide (chromium (Cr)), specularite (Fe), yttrium Al garnet (yttrium (Y), zircon (Zr), niobium metal (Nb), tantalum metal (Ta), tungsten metal (W), cubic zirconia (hafnium (Hf)), apatite (P), fluorite (fluoride (F)), halite (chlorine (Cl)) and celestine (sulfur (S) and Sr). The methodology for the analysis of REE followed that of ref. [38]. The detection limits of Zr, REE and Y vary from about 100 ppm to about 300 ppm, whereas those for major elements are 60–120 ppm. Representative analysis results for phosphate and Zr-bearing minerals are listed in Supplementary Tables 1, 2, respectively.

## SIMS analyses

The target selection strategy was to identify phases that would contain initial Pb and those containing radiogenic Pb generated from in situ decay of U since crystallization of the sample (for example, phosphates and Zr-bearing phases). The distinct Pb isotopic ratios yielded by these two types of phase help to populate the isochron and calculate a precise date. The target areas within these phases were selected to be large enough to accommodate a SIMS analytical spot (in this case, the smallest spot used was less than 3 µm). The Pb isotopic compositions (complete dataset presented in Supplementary Table 3) of the phases were determined over three analytical sessions using a CAMECA IMS 1280HR ion microprobe. The mounts with candidate minerals were cleaned with a fine (0.25 µm) diamond paste and ethanol to remove the carbon coating before adding a roughly 20-nm gold coating.

The first session focused on the Pb-isotope measurement of Zr-bearing minerals. A Gaussian illumination mode was used to focus a primary beam of $^{16}O^-$ to a size of less than 3 µm (about 2.8 µm) (Fig. 1, Extended Data Fig. 2), with an accelerated potential of −13 kV. The beam size can be kept unchanged for a long usage time and intensities were around 250–200 pA. The primary beam setting is described in detail in ref. [39]. The multi-collector mode with five electron multipliers with low noise (less than 0.001 cps, especially for L2) was used to measure $^{204}Pb^+$ (L2), $^{206}Pb^+$ (L1), $^{207}Pb^+$ (C), $^{208}Pb^+$ (H1) and $^{96}Zr_2^{16}O_2^+$ (H2). The methodology is similar to that outlined in ref. [40]. Exit slit 3 was used, with a mass resolving power (MRP) of 8,000 (50% peak height). Before analysis, a primary beam of $^{16}O^-$ with an intensity of 10 nA was used for 120 s of pre-sputtering. The ion images with $^{96}Zr_2^{16}O_2^+$ and Pb isotopes on a 25 µm × 25-µm area were used to precisely locate the target minerals. The signal of $^{206}Pb$ was used for peak-centring reference. Each measurement consisted of 4 s × 80 cycles, with a total analytical time of about 10 min. High-purity oxygen gas was leaked onto the sample surface to enhance the $Pb^+$ yield to more than 15 cps ppm$^{-1}$ nA$^{-1}$ by using a $O^-$ primary beam according to the M257 zircon standard (561 Ma, 840 ppm U, ref. [41]). NIST610 glass and Phalaborwa baddeleyite standard ($^{207}Pb^*/^{206}Pb^* = 0.1272$, ref. [42]) were used to calibrate the relative yield of different electron multipliers and evaluate the external reproducibility. On the basis of 21 analyses on NIST610 glass under the same analytical conditions, the $^{207}Pb/^{206}Pb$ measurements have a relative standard deviation (1 r.s.d.) of 0.66% with $^{207}Pb$ intensity averaged at 127 cps. The possible SIMS instrumental mass fractionation of Pb isotopes around 0.2% (ref. [43]) was propagated to the uncertainty of single-spot $^{207}Pb/^{206}Pb$ analysis.

The second session focused on the Pb-isotope measurement of phosphates. A Gaussian illumination mode was used to focus a primary beam of $^{16}O_2^-$ to a roughly 8-µm size with the intensity kept around 2.5 nA. Ion images with $Ca_2PO_3^+$ on a 30 µm × 30-µm area were used to precisely locate the target minerals. The methodology is similar to that outlined in ref. [44]. Exit slit 3 was used, with an MRP of 8,000 (50% peak height). The dynamic multi-collector mode was used to measure $^{204}Pb^+$ (L2), $^{206}Pb^+$ (L1) and $^{207}Pb^+$ (C) in the first step with 60-s counting time, and $^{238}U$ (L1), $^{232}ThO^+$ (H1) and $^{238}UO^+$ (H2) in the second step with 4-s counting time. Each measurement consisted of 10 cycles, with a total analytical time of about 15 min including 2 min of pre-sputtering. NIST610 glass and Phalaborwa baddeleyite ($^{207}Pb^*/^{206}Pb^* = 0.1272$; ref. [42]) were used to calibrate the relative yield of different electron multipliers and evaluate the external reproducibility. On the basis of 20 analyses on Phalaborwa baddeleyite under the same analytical conditions, the $^{207}Pb/^{206}Pb$ measurements show 1 r.s.d. of 0.2% with $^{207}Pb$ intensity averaged at 970 cps.

The third session focused on the initial Pb composition test on the essential minerals (mainly plagioclase and pyroxene) and matrix. A Köhler illumination mode was used to produce a primary beam of about 30 nA $O_2^-$ to a roughly 30-µm size. Before each measurement, an area of 25 µm around the spot location was raster-scanned for 120 s to remove the gold coating and minimize possible surface contamination. The multi-collector mode with five electron multipliers was used to measure $^{204}Pb^+$ (L2), $^{206}Pb^+$ (L1), $^{207}Pb^+$ (C) and $^{208}Pb^+$ (H1) in the first step with 60-s counting time, and $^{232}ThO^+$ (H1) and $^{238}UO^+$ (H2) in the second step with 4-s counting time. Exit slit 3 was used, with an MRP of 8,000 (50% peak height), sufficient to resolve Pb from known molecular interferences. Each measurement consisted of 10 cycles. On the basis of 28 analyses on NIST614 glass under the same analytical conditions, the $^{207}Pb/^{206}Pb$ measurements show 1 r.s.d. of 0.68% with $^{207}Pb$ intensity averaged at 120 cps.

The background counts for each channel were measured at regular intervals during each session by using deflector and aperture settings that effectively blank both primary and any residual secondary beams. The average background values are reported in Extended Data Table 2. The number of SIMS Pb isotope analyses for each type of basalt clast used in this study are summarized in Extended Data Table 1.

## SIMS data processing

The data were processed using in-house SIMS data reduction spreadsheets and the Excel add-in Isoplot (version 4.15; ref. [45]). The Pb–Pb isochrons were constructed following the method first established by ref. [46], then applied to SIMS by ref. [6] and further expressed and

refined by refs. [7,10]. In brief, the Pb isotopic compositions measured in each sample are interpreted as representing a mixture between three main components: (1) initial Pb present in the basaltic melt when it crystallized; (2) radiogenic Pb formed by the decay of U in the basalt after crystallization ;and (3) terrestrial contamination. These end-member components define a triangular array of points on a plot of $^{207}Pb/^{206}Pb$ versus $^{204}Pb/^{206}Pb$ (Extended Data Fig. 3). The values with the highest $^{207}Pb/^{206}Pb$ ratios, those at the top of the triangular array, provide an estimate of the lowest possible value for the initial Pb composition of the sample. The radiogenic Pb component is then located where $^{204}Pb/^{206}Pb = 0$. Finally, given the radiogenic Pb isotopic compositions associated with the Moon relative to those found on Earth, the terrestrial contamination end-member will have the highest $^{204}Pb/^{206}Pb$ ratios. Any obvious cracks or voids in the sample would be likely places for terrestrial contamination to accumulate during sample polishing and cleaning procedures. Despite the efforts made to avoid such regions, it was difficult to ensure that the SIMS spots did not overlap with small invisible cracks, or did not depth-profile into such features lying just below the original surface of the sample. Furthermore, given the low Pb concentrations in many of the analysed phases (particularly those with lower radiogenic Pb isotope compositions), these analyses are particularly susceptible to even low levels of terrestrial contamination. On the basis of these assumptions, the bounding edge on the left side of the triangle, between the initial and radiogenic lunar Pb compositions, forms an isochron, which can be determined by iteratively filtering the data to yield the steepest statistically significant weighted regression. The $^{207}Pb/^{206}Pb$ of initial Pb was estimated by spots with the highest $^{207}Pb/^{206}Pb$ and near-zero $^{238}UO^+/^{208}Pb^+$ (Extended Data Fig. 4), following the procedures of refs. [7,10].

The Pb concentrations were estimated based on the Pb yield assuming that Zr-bearing minerals have comparable Pb yield with that in zircon (that is, 15 cps ppm$^{-1}$ nA$^{-1}$ with a primary beam of O$^-$). Baddeleyite, tranquillityite and zirconolite grains have U contents of approximately 100–4,000 ppm, 600–7,000 ppm and 800–4,600 ppm, respectively, whereas the phosphate grains have lower U contents of approximately 12–150 ppm estimated from UO$^+$ yield based on Durango apatite.

## Data availability

All data generated or analysed during this study are available in Earth-Chem Library at https://doi.org/10.26022/IEDA/112085. Source data are provided with this paper.

## Code availability

No code is used in this study.

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

**Acknowledgements** We thank all the staff of China's Chang'e Lunar Exploration Project for their hard work in returning lunar samples. The samples studied in this work were allocated by the China National Space Administration. We thank R. Mitchell for constructive comments and editing the manuscript; W. Yang, Y. Chen, H.-J. Hui and H.-C. Tian for helpful discussion; and C. Sun for logistical support. This study was funded by the Key Research Program of the Chinese Academy of Sciences (ZDBS-SSW-JSC007-13), the Institute of Geology and Geophysics, Chinese Academy of Sciences (IGGCAS-202101), the National Natural Science Foundation of China (41773044) and the pre-research project on Civil Aerospace Technologies of China National Space Administration (grant number D020203).

**Author contributions** X.-H.L. and C.L. conceived and supervised this project. Q.-L.L., Q.Z., and Z.X. wrote the manuscript with input of X.-H.L., C.L., Y.L. and F.-Y.W. H.-X.M. and J.L. conducted sample mounting. Q.Z., J.-H.L., X.T., S.G. and J.-Y.Y. performed SEM and EPMA analyses. Q.-L.L., Y.L., G.-Q.T. and Q.Z. conducted SIMS analyses, data processing and interpretation. Z.O. and F.-Y.W. contributed scientific background and geological context.

**Competing interests** The authors declare no competing interests.

**Additional information**
**Correspondence and requests for materials** should be addressed to Chunlai Li or Xian-Hua Li.

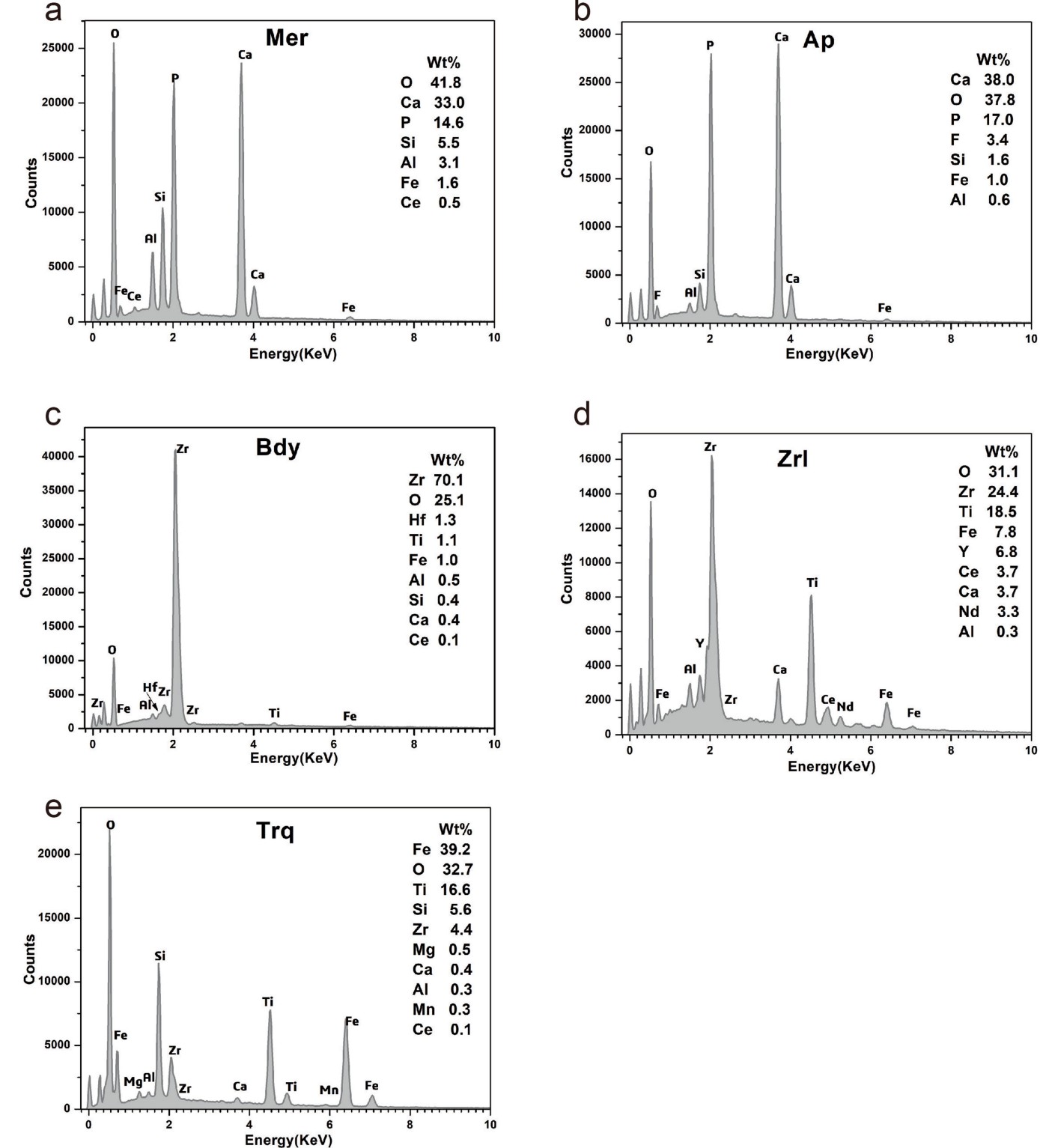

**Extended Data Fig. 1 | Plots of the representative energy dispersive X-ray spectrum for dated minerals.** Element concentrations detected by EDS are also shown for comparison. Mer, merrillite; Ap, apatite; Bdy, baddeleyite; Zrl, zirconolite; Trq, tranquillityite.

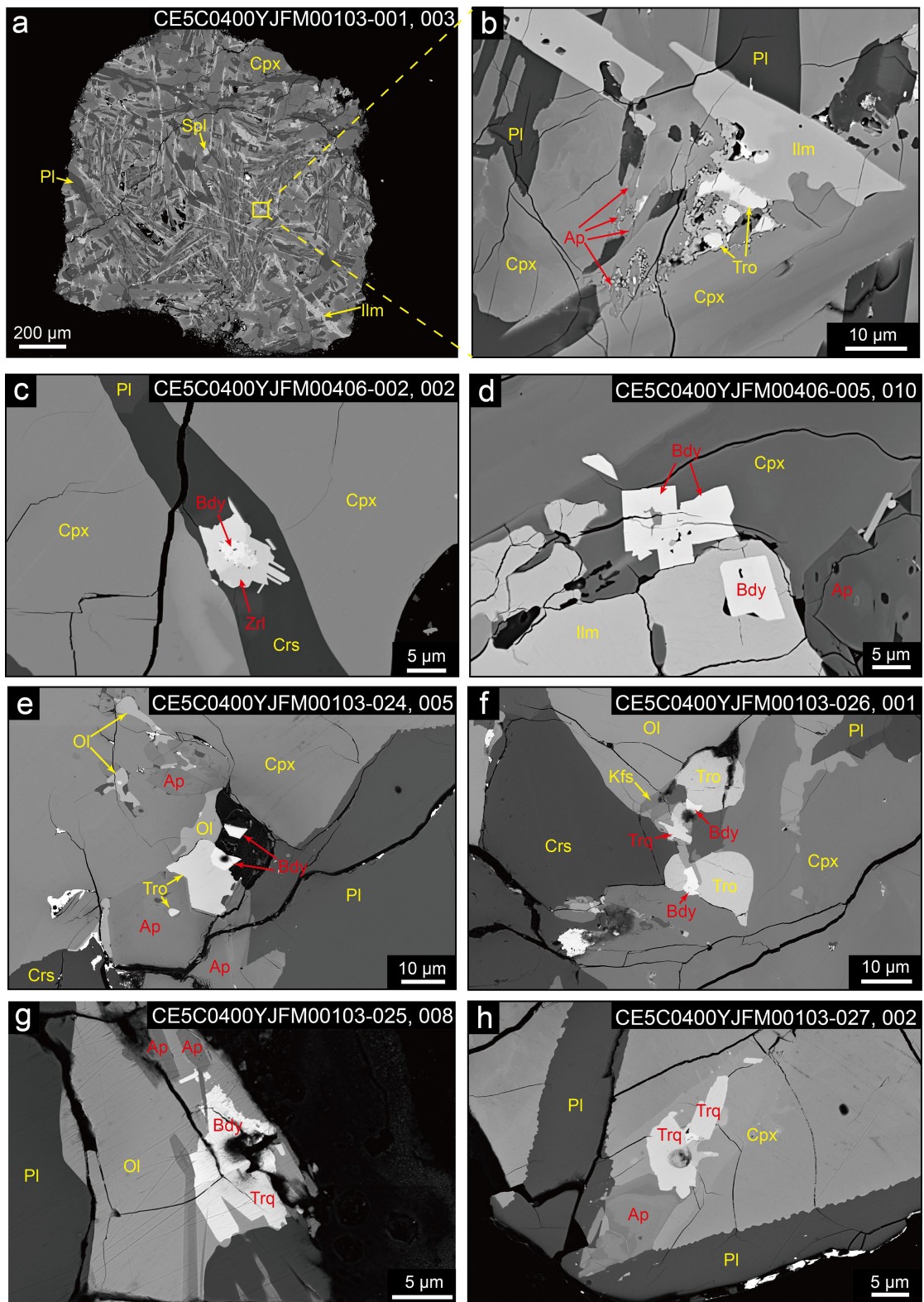

**Extended Data Fig. 2 | Microtextures of dated minerals in the Chang'e-5 basalt clasts. a**, **b**, A porphyritic basalt clast that contains fine-grained (2 × 10 μm) apatite grains. The yellow rectangle in **a** is expanded in **b**. The apatite occurs as inclusions in Fe-rich clinopyroxene and is surrounded by tiny ilmenite aggregates. **c**, In a poikilitic clast, baddeleyite is rimmed by zirconolite. **d**, Square-shaped baddeleyite inclusions in clinopyroxene and ilmenite from an equigranular clast. Hexagonal apatite exhibits an intergranular phase between clinopyroxene and ilmenite. **e**, Euhedral–subhedral apatite and baddeleyite show an equilibrium texture with Fe-rich olivine (Fo < 10), clinopyroxene, and troilite. **f**, Baddeleyite and tranquillityite show intergrowths along the margins of clinopyroxene and Fe-rich olivine. **g**, In an equigranular clast, baddeleyite, tranquillityite and apatite are intergrown with Fe-rich olivine. **h**, Tranquillityite and apatite intergrown crystals in a poikilitic clast. Blurry pits in Zr-bearing minerals are the analytical spots from SIMS. Bdy, baddeleyite; Zrl, zirconolite; Trq, tranquillityite; Mer, merrillite; Ap, Apatite; Cpx, clinopyroxene; Ol, olivine; Pl, plagioclase; Kfs, K-feldspar; Ilm, ilmenite; Crs, cristobalite; Tro, troilite; Spl, spinel.

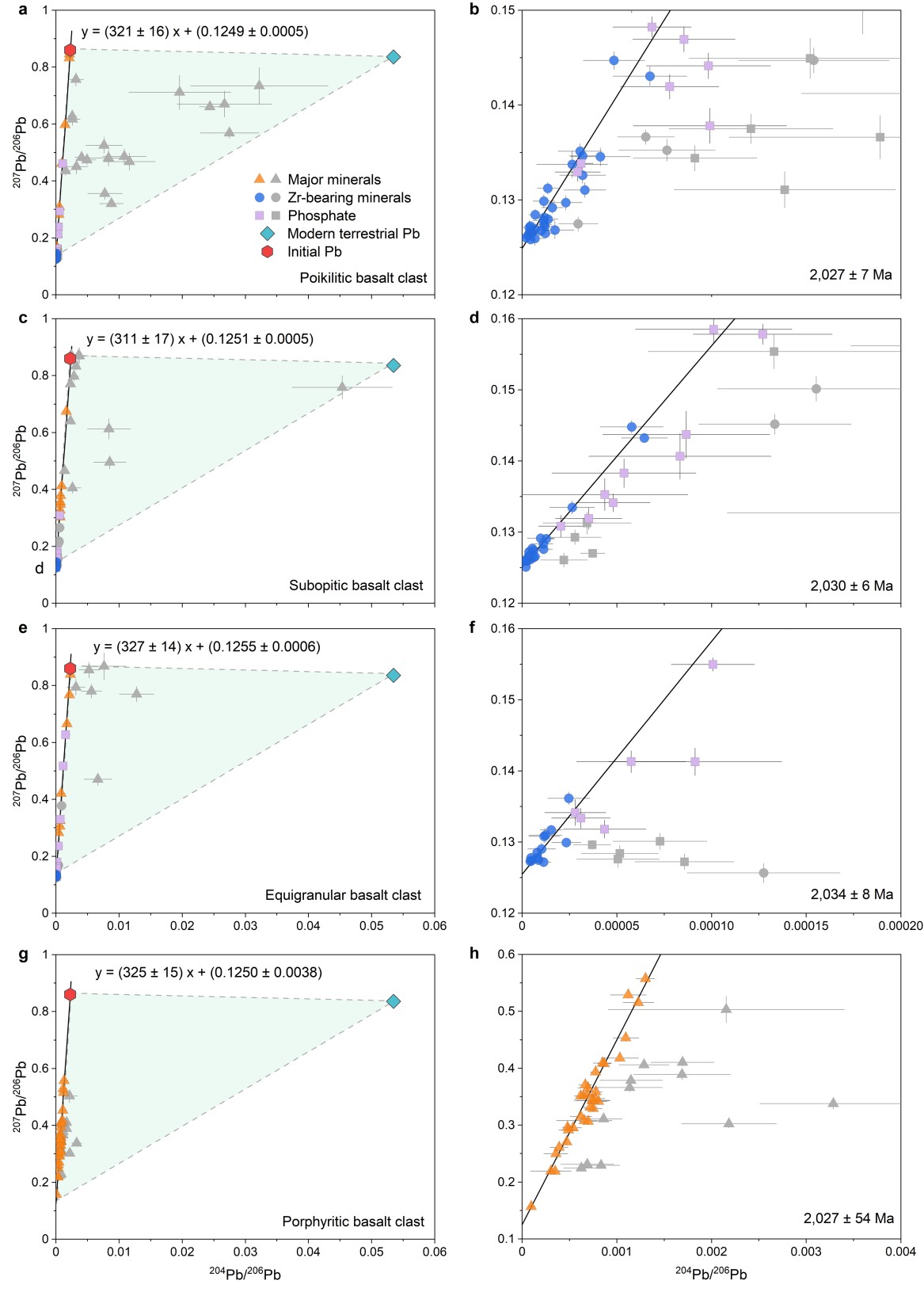

**Extended Data Fig. 3 | Pb–Pb isochrons for the Chang'e-5 basalts with different textures.** The left four plots (**a**, **c**, **e** and **g**) show the data from basalt clasts with poikilitic, subophitic, equigranular and porphyritic textures, respectively. The equations of isochrons (black lines) are shown on the top. The right four plots (**b**, **d**, **f** and **h**) are the enlarged lowest portions of the isochrons highlighting the measurements with low $^{204}Pb/^{206}Pb$. The red hexagon represents the initial Pb determined from the integrated isochron, but is not included in each separated isochron. The triangle areas represent the mixing trend among the initial Pb component, the radiogenic Pb, and current terrestrial Pb composition. Outliers excluded from the calculation of the isochron regression are shown in grey while those data used for the leftmost isochron are shown in colour. Error bars represent 1σ standard error. The uncertainties for the isochron dates are quoted at the 95% confidence level.

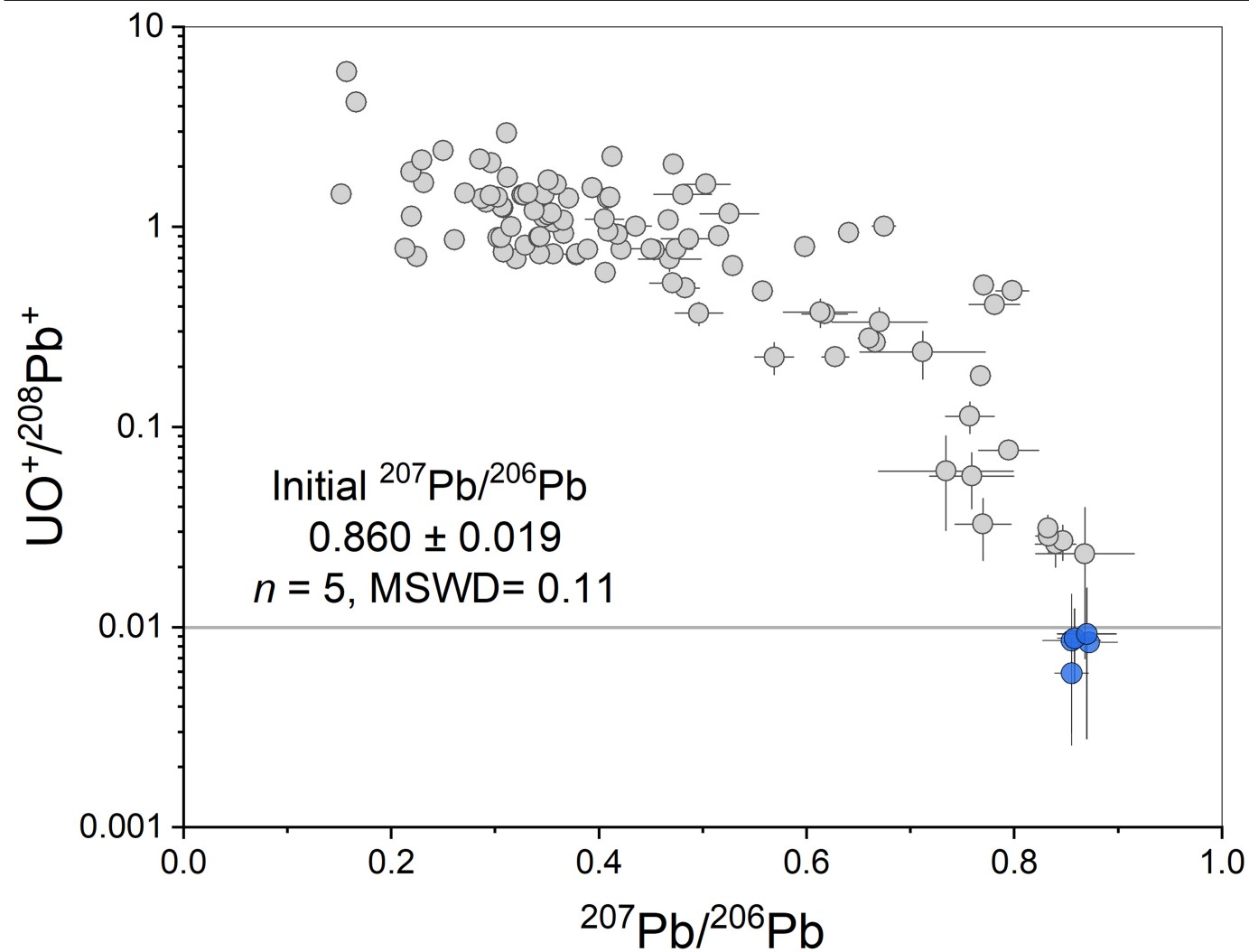

**Extended Data Fig. 4 | Plot of UO⁺/²⁰⁸Pb⁺ versus ²⁰⁷Pb/²⁰⁶Pb for points within the analysed main rock-forming minerals.** Five blue dots highlight the points with UO⁺/²⁰⁸Pb⁺ < 0.01 (the grey line), that most likely represent the best estimate for the initial Pb composition. Note that the measured UO⁺/²⁰⁸Pb⁺ ratios are simply used here to provide an indication of the U/Pb ratios. Error bars represent 1σ standard error.

**a**

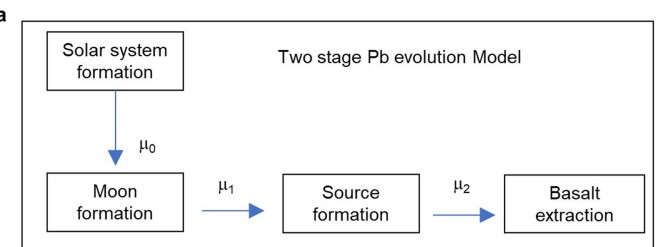

Two stage Pb evolution Model

Solar system formation → (μ₀) Moon formation → (μ₁) Source formation → (μ₂) Basalt extraction

**b**

Equations used in two stage Pb evolution Model

$$\frac{^{206}Pb}{^{204}Pb} = \frac{^{206}Pb}{^{204}Pb}_{moon} + \mu_1\left(e^{\lambda_1 T} - e^{\lambda_1 t_1}\right) + \mu_2\left(e^{\lambda_1 t_1} - e^{\lambda_1 t_2}\right)$$

$$\frac{^{207}Pb}{^{204}Pb} = \frac{^{207}Pb}{^{204}Pb}_{moon} + \frac{\mu_1\left(e^{\lambda_2 T} - e^{\lambda_2 t_1}\right) + \mu_2\left(e^{\lambda_2 t_1} - e^{\lambda_2 t_2}\right)}{137.818}$$

**c**

| Basic Parameters | $\lambda_1$ | $\lambda_2$ | $^{235}U/^{238}U$ |
|---|---|---|---|
| | 1.55125E-10 | 9.8485E-10 | 0.007256 |

**d**

| Known values | Age (Ma) | $^{206}Pb/^{204}Pb$ | $^{207}Pb/^{204}Pb$ | $^{204}Pb/^{206}Pb$ | $^{207}Pb/^{206}Pb$ | $\mu$ ($^{238}U/^{204}Pb$) |
|---|---|---|---|---|---|---|
| Solar system | 4567 | 9.307 | 10.294 | 0.1074 | 1.106 | 8 |
| Moon formation | 4500 | 9.475 | 10.63 | 0.1055 | 1.122 | 462±46 |
| Sample extraction | 2030±4 | 439 ± 21 | 377 ± 19 | 0.00228 ± 0.00011 | 0.860 ± 0.019 | |

**e**

| | | | $\mu_2$ values with different model ages | | | |
|---|---|---|---|---|---|---|
| Age (Ma) | $^{206}Pb/^{204}Pb$ | $^{207}Pb/^{204}Pb$ | $\mu_2$ from $^{238}U$-$^{206}Pb$ | Error | $\mu_2$ from $^{235}U$-$^{207}Pb$ | Error |
| 4420 | 20.88 | 31.89 | 679 | 36 | 677 | 40 |
| 4400 | 23.71 | 36.95 | 681 | 37 | 681 | 42 |
| 4380 | 26.53 | 41.91 | 684 | 38 | 686 | 44 |
| 4376 ± 18 | 27.09 | 42.89 | 684 | 40 | 687 | 46 |
| 4360 | 29.34 | 46.77 | 686 | 39 | 691 | 46 |
| 4340 | 32.14 | 51.54 | 688 | 40 | 696 | 48 |
| 4320 | 34.93 | 56.22 | 691 | 41 | 702 | 50 |
| 4300 | 37.72 | 60.80 | 693 | 42 | 707 | 52 |
| 3350 | 160.5 | 200.9 | 893 | 117 | 1233 | 261 |

**Extended Data Fig. 5 | The two-stage Pb evolution model and $\mu$ value calculation procedures. a**, The schematic diagram of two-stage Pb evolution model[6]. **b**, The equations used in the two-stage Pb evolution model. **c**, The basic parameters used, including the decay constant of $^{238}U$ and $^{235}U$[47] and $^{235}U/^{238}U$ ratio[48]. **d**, The summarized parameters from previous work[6,49] and this study. **e**, The calculated $\mu$ value based on different modelled ages.

**Extended Data Table 1 | Summary of SIMS Pb isotope analyses for each type of Chang'e-5 basalt clast**

|  |  | Poikilitic | Subophitic | Equigranular | Porphyritic | Total |
|---|---|---|---|---|---|---|
| Basalt clast numbers |  | 17 | 18 | 10 | 2 | 47 |
| Zr-bearing minerals grains | Baddeleyite | 17 | 12 | 6 | 0 | 35 |
|  | Tranquillityite | 3 | 3 | 1 | 0 | 7 |
|  | Zirconolite | 4 | 2 | 3 | 0 | 9 |
| Phosphate grains |  | 15 | 15 | 10 | 0 | 40 |
| Plagioclase + pyroxene + matrix |  | 27 | 22 | 13 | 44 | 106 |
| Isochron Age (Ma) |  | 2,027 ± 7 | 2,030 ± 6 | 2,034 ± 8 | 2,027 ± 54 | 2,030 ± 4 |

**Extended Data Table 2 | Background measurements for electron multiplier collectors**

| EM collectors | L2 | | L1 | | C | | H1 | |
|---|---|---|---|---|---|---|---|---|
| used for | $^{204}$Pb (cps) | ± | $^{206}$Pb (cps) | ± | $^{207}$Pb (cps) | ± | $^{208}$Pb (cps) | ± |
| Session 1 (n=20) | 0.00071 | 0.00134 | 0.00412 | 0.00315 | 0.00355 | 0.00141 | 0.00071 | 0.00109 |
| Session 2 (n=30) | 0.00057 | 0.00145 | 0.00055 | 0.00145 | 0.00046 | 0.00104 | 0.00037 | 0.00104 |
| Session 3 (n=25) | 0.00087 | 0.00143 | 0.00050 | 0.00116 | 0.00063 | 0.00156 | 0.00050 | 0.00147 |

The values presented here are the average values of the measured backgrounds for each session.
The errors are 1$\sigma$ standard deviations of the average values.