## [Peer Review File · Nature]

Manuscript Title: Two billion-year-old volcanism on the Moon from Chang'E-5 basalts

Reviewer Comments & Author Rebuttals

Reviewer Reports on the Initial Version:

Referee #1:

This is an important manuscript for several reasons: the manuscript provides chronology information for the newly returned Chang'E-5 basalts, these are the youngest basalts thus far identified from the Moon, the young age of the basalts provides a pivotal calibration point to the crater-counting chronology for the Moon and perhaps the inner Solar System., and it provides further evidence for basalts produced from melting of a KREEP-depleted source within the PKT region of the Moon. The has some minor grammatical issues that need to be corrected. For example lines such as 31-34, 48-51 are awkward and need to be corrected.

Comments on the content of the manuscript.

BSE images are beautiful and exhibit the textural nature of the basalts and individual mineral phases.

Line 71: What is the nature of the glasses. Are they impact, volcanic, or both?

Line 91-92: Is there evidence that these fragments represent the sample flow or period of magmatism?

Line 111. State the character of the primary beam (e.g., O-, Cs+).

Line 117. Be more specific than "rock-forming minerals"

Line 119. Ma rather than Ga?

Line 121. Provide evidence for the statement "appear to have crystallized synchronously". Errors associated with the Pb-Pb ages are millions of years.

Lines 131-134. This is a subtle point that needs to be rewritten. The returned samples are younger than any lunar mare basalt sample. However, most lunar scientists realized that mare magmatism was younger than the youngest samples. The young ages of the basalts returned by the Chang'E-5 mission were expected.

Line 157. This model age from the literature is different from the model age determined for these basalts in a associated manuscript (Non-KREEP origin for Chang'E-5 basalt in the Procellarum KREEP Terrane". Why not use the made age measured on these samples? What is the μ value if the older model age is used? What is the interpretation of the model age? What is the origin for the difference in μ values between KREEP and the sources for the mare basalts?

Lines 161-162. Again a subtlety in the interpretation. Heat generated by KREEP may have heated the mantle without being in the sources for these basalts?

Lines 180-182. Not sure what this means. Clearly, extending the crater counting among the Apollo sites provides evidence for the reliability of crater counting on the Moon over a restricted period of time.

Charles "Chip" Shearer

This manuscript documents the age and Pb initial isotopic composition of basaltic clasts from lunar soils recently collected by the Chang'E 5 sample return mission. These data provide unprecedented new insight in the chronology of the lunar magmatic activity based on exceptional samples. Furthermore, using their new age, the authors provide further constraints on calibrating the crater counting method.

The manuscript is well written and well organised. The data are solid and the analytical methods used to acquire them as well as data processing protocols are sound. In this latter aspect, the authors did a very good job: they have assessed the terrestrial Pb contamination in the analyses and filtered the dataset accordingly. Failing to do this procedure was a common mistake in the past that crippled many studies based on U-Pb or Pb-Pb dating techniques yet still seen in many recent papers dealing with lunar rocks. This makes this study sound and the ages reliable then worth publishing in Nature.

Nevertheless, this manuscript is not perfect and needs some improvement.

Specifically, I noticed one potential issue concerning the calculation of the Pb initial isotopic composition. This section of the paper is confusing in particular when reading the captions of figure 2. As it stands (as I understand it), the caption and the text suggest that the Pb initial $^{207}\text{Pb}/^{206}\text{Pb}$ ratio was calculated from 6 data. It is not mentioned if the calculation is a weighted average. In this case, the calculated values should be given with relevant statistical parameters and the corresponding plots could be provided in supplementary material.

The initial $^{204}\text{Pb}/^{206}\text{Pb}$ ratio was calculated using a 2-stage Pb evolution model and an evolution curve with a μ value of 1000; this evolution curve being the best fit for intersecting the isochron where are located the data with the highest $^{207}\text{Pb}/^{206}\text{Pb}$ ratios (and then, the calculated average initial $^{207}\text{Pb}/^{206}\text{Pb}$ ratio). This μ value must have been estimated before the calculation of the initial $^{204}\text{Pb}/^{206}\text{Pb}$ ratio. But once, this initial $^{204}\text{Pb}/^{206}\text{Pb}$ ratio is calculated, it is used in turn to calculate the μ value of the mantle source, as seen in Source data Table 2. If this is correct, it looks like there is an issue of circular argument regarding the calculation of the initial $^{204}\text{Pb}/^{206}\text{Pb}$ ratio and μ value of the mantle source. The authors need to either better explain the procedure leading to the calculation of the Pb initial isotopic composition or revise this procedure to avoid this circular argument issue. Perhaps, using a weighted average for the calculation of the $^{204}\text{Pb}/^{206}\text{Pb}$ ratio is the best way despite this could yield a less precise average ratio then less precise μ value. This is probably my main concern about this manuscript.

I think the manuscript is a bit unbalanced: the discussion about the Pb initial isotopic composition of the Moon is quite small (L161-L168) in comparison to the discussion about calibrating the crater counting method (L175-L199). I think this is related to the fact that the former is a bit underdeveloped while the importance of the latter is a bit overstressed.

Calibrating the crater counting method is important but I would say that it is a secondary issue compared to the determination of Pb isotopic composition of lunar basalts then identification of possible mantle reservoirs (including KREEP). The crater counting method is a relative chronology method which is useful as we don't have enough samples to assess the period of activity of lunar magmatism. Its application on other astronomical bodies is in fact limited as we still need a representative set of samples to be calibrated this method (samples we do not have) and of course, a large planetary surface covered by craters. As soon as we will have a large set of samples; this method will most likely be obsolete.

I would encourage the authors to further discuss the implications of having basalts derived from a source with relatively low μ values sampled in Procellarum terrane where basalts are expected to be derived from source with high μ values or try to explain why basalts tend to show minor or none contribution of a KREEP component as they are getting younger.

I also noted few minor issues that listed below:

- Abstract: the mention of the crater counting method and its calibration come on Line 4 while the aspect of Pb isotopic composition of mantle reservoirs is mentioned at the end of the "results" section of the abstract close to the end of it. I would place this aspect of mantle reservoir more at the beginning of the abstract and mention the crater counting calibration at the end of the "results" section. By doing this, the abstract would follow the layout of the manuscript where the discussion about crater counting method is at the end of the paper.
- Source data table 1 needs some improvement. The caption mentions that "shadowed data" were not used for the isochron construction but the data are either in black font or red font so the term "shadowed" seems inadequate. A column with the actual $^{238}\text{UO}^+ / ^{208}\text{Pb}^+$ should be included in this table for the plagioclase analyses. There is column with the ^{208}Pb counts for phosphates: this seems rather weird but is any reason for that? Finally, some plagioclase data are outlined as used in the calculation of the Pb initial isotopic ratios but among these data; some of them are in black font while others are in red font suggesting that some data rejected for the isochron construction might have been used in the initial ratio calculation. Actually, according to this table, seven plagioclase data seem to have been used in the calculation of the Pb initial ratio instead of 6 as mentioned in the text. Finally, two data among these plagioclase analyses used

to the calculation of the initials (again, if I read correctly the table caption), have U/Pb ratio higher than 0.01. According to the text, these data should have been discarded for the initials calculation. All these little issues should be either fixed.

- In the supplementary material Tables 1 and 2 (Electron microprobe data), this is a minor detail but when recalculating the totals of the analyses, I found a discrepancy of 0.02-0.01% between my calculation and the totals indicated in the paper.
- In the supplementary material Table 3, it is not clear why there are two lines for the background noise. Does it mean that these background measurements correspond to two different analytical sessions? In the supplementary material, three analytical sessions are mentioned. This needs to be clarified.

Here below are line-by-line comments on specific issues I spotted across the manuscript:

- L72-L76: I don't really understand this section: at the beginning of the sentence, the authors say they picked clasts from 2 lunar soils but the next part of the same sentence, I understand they mounted another soil sample in epoxy as well as 4 basaltic clasts in in 2 other epoxy mounts. To be mounted in epoxy; the grains in these 3 mounts must have been picked up from bulk soil. This is even more confusing when reaching line 91 where it is said that 47 basaltic clasts were investigated for Pb-Pb dating. The reader needs to know how many different samples scooped from the surface were investigated, how many grains were picked from these samples and how many were mounted in each epoxy mount and if there is a lithology sorting in the different epoxy mounts
- L92: Do the authors have an idea of the bulk chemistry of the clasts: for the EDS maps, a rough estimate of the chemistry is possible. the bottom line is: are the clasts low-Ti, High-Ti, high-Al? or even KREEP?
- L122-123: I am not comfortable with this: the cited reference is a submitted manuscript. I think the authors made their points by saying that the different clasts formed at the same time.
- L133-134: The authors could be more explicit here and give a hint about their opinion about what are these new constraints
- L137-138: It looks like this evolution curve has a μ value of 1000. Is that correct? I do not really understand the meaning of the ticks on this curve and of "4376-2030 Ma": is this the age span of the second stage of Pb evolution?

- L138-140: This sounds like the authors found the best fitting evolution curve which has a μ value of 1000. This best fitting curve being those intersecting the isochron at its extremity (where plot the data with the highest $^{207}/^{206}\text{Pb}$ ratios). Following this, the authors determine the Pb initial isotopic ratios.
- L148: "have been calculated" or "can be calculated"?
- L150: It looks like some data in this table used in the calculation of the initial $^{207}/^{206}$ ratio have $^{238}\text{UO}+ / ^{208}\text{Pb}+$ ratio higher than 0.01.
- L152-155: This is a weighted average? if so, say it. If the authors do not want to use a weighted average calculation for determination of $^{204}/^{206}$ initial ratio, why not using the isochron to determine the $^{204}/^{206}$ ratio: the authors have already the $^{207}/^{206}$ ratio, it seems straightforward
- L156-158: How this μ value is calculated needs some clarifications: this value cannot be calculated using a $^{204}/^{206}$ ratio determined from the intersect between the isochron and an evolution curve with an estimated μ value representing the best fit for such intersect: it is a circular argument issue....
- L160-163: This statement is not clear and might be borderline overinterpretation: the data tend to suggest a limited if any involvement of a KREEP component in the geochemical characteristics of the source of the basalts but the authors seem to assume that this is a contribution of a KREEP reservoir in the source of these lunar basalts. This has to be explained or clarified. The authors can outline that no or very little KREEP contribution is quite unusual for basalts from Procellarum terrane which is the locus of KREEP-flavoured basalts. The authors seem to suggest that contamination of Low-Ti basalts en-route to the surface by material with KREEP chemical signature but it is not clear how they reach this conclusion. This has to be explained/demonstrated. Furthermore, the authors might have also to consider how their data fit or do not fit with the hypothesis expressed by Merle et al. (2020) suggesting that the low-Ti basalts seem to be more contaminated by a KREEP component as they are younger.
- L165-166: The authors could remove this sentence as they cannot rely on the results/interpretations from this manuscript as long as it is not published.
- It is not very critical anyway.
- Figure 3: There are also data of low-Ti basalts from lunar meteorites. There is no obvious reasons to not use these data for comparison.
- L182: if this is a model age, it cannot be "absolute".

- L198-199: Please explain. It is obvious for the Moon but for other planets, it is not that clear: except Mars, we don't have much samples from other planets...
- L302: Very minor detail: it is an EDS detector: EDS is energy dispersive x-ray spectroscopy. So it is spectroscopy not spectrometry. Strictly speaking...
- L315-316: replace "obtained by" by "processed with".
- L323-324: please give detection limit for major elements.
- L330-332: this sentence could be improved: "the distinct Pb isotopic ratios yielded by these two types of phases help populating the isochron and calculate a precise date".
- L339-341: it looks like this is expressed in the next few lines (L342-L354). The authors could slightly rephrase to make it a bit more straightforward.
- L339: replace "is" by "focused on"
- L349: replace "was" by "were"
- L350: replace "position" by "localise"
- L362-369: Mass resolution and standards used as well as the average values of their isotopic ratios and related analytical precision should be indicated here as for the first analytical session.
- L370-379: standards? Related average value and precision should be also provided.
- L380-381: there are 2 lines with 20 and 30 background measurements in this table. Therefore, it is not clear if these 2 sets correspond to only 2 analytical sessions. This would mean that backgrounds for one analytical session are missing. Indicate clearly the corresponding analytical session for the different sets of background analyses in this table.
- L389-394: Here the authors should mentioned that the method was first established by Connelly et al., 2012 (Science, 338, P651-655) then applied to SIMS by Snape et al., 2016 and further expressed and refined by Snape et al., 2019 and Merle et al., 2020. Indeed L390-L393 were stated first by Connelly et al., 2012.
- L407-410: Here cite Snape et al., 2019 and Merle et al., 2020/ these 2 papers clearly expressed this procedure.
- L410-411: Did the authors calculate a weighted average for these values?

As a conclusion, there are only few issues that are relatively easy to fix but this requires an extra bit of work before publication.

Best wishes,

Renaud Merle

Author Rebuttals to Initial Comments:

Reviewer #1 Evaluations:

This is an important manuscript for several reasons: the manuscript provides chronology information for the newly returned Chang'E-5 basalts, these are the youngest basalts thus far identified from the Moon, the young age of the basalts provides a pivotal calibration point to the crater-counting chronology for the Moon and perhaps the inner Solar System., and it provides further evidence for basalts produced from melting of a KREEP-depleted source within the PKT region of the Moon.

We appreciate the reviewer's positive evaluation on the scientific value of this work.

The has some minor grammatical issues that need to be corrected. For example lines such as 31-34, 48-51 are awkward and need to be corrected.

All minor grammatical issues have been corrected.

All the suggestions below have been accepted, and changes are integrated into this revision.

Comments on the content of the manuscript.

BSE images are beautiful and exhibit the textural nature of the basalts and individual mineral phases.

Thanks.

Line 71: What is the nature of the glasses. Are they impact, volcanic, or both?

Based on our preliminary observations and investigation until now, most glasses in Chang'E-5 returned samples (that we obtained) are impact glasses, with only a few grains likely being volcanic glasses. The nature of glasses is beyond the scope of this study; thus, we did not describe it in detail. Several research teams are currently working on the glasses.

Line 91-92: Is there evidence that these fragments represent the sample flow or period of magmatism?

In this paper, we did not provide evidence that these fragments represent the same flow or magmatic episode. Four isochrons from basalt fragments with different textures show consistent y -intercepts (corresponding to ages) and slopes within uncertainties, indicating both their identical ages and derivation from most likely the same source. Another companion manuscript "Non-KREEP origin for Chang'E-5 basalts in the Procellarum KREEP Terrane" provides more clues to the derivation from the same flow.

Line 111. State the character of the primary beam (e.g., O⁻, Cs⁺).

It is O⁻. We revised accordingly.

Line 117. Be more specific than "rock-forming minerals"

Accepted. We list the analyzed rock-forming minerals as plagioclase and pyroxene in the revision.

Line 119. Ma rather than Ga?

We apologize for this typo. It is corrected as “Ma”.

Line 121. Provide evidence for the statement “appear to have crystallized synchronously”. Errors associated with the Pb-Pb ages are millions of years.

Our results suggest that the crystallisation ages for Chang'E-5 basalt clasts with different textures are indistinguishable within analytical uncertainties, but that does not mean that they are strictly “synchronous”. We delete this vague statement.

Lines 131-134. This is a subtle point that needs to be rewritten. The returned samples are younger than any lunar mare basalt sample. However, most lunar scientists realized that mare magmatism was younger than the youngest samples. The young ages of the basalts returned by the Chang'E-5 mission were expected.

Thanks for your suggestion. We revise this sentence as: *“Therefore, this study provides the first conclusive evidence that magmatic activity on the Moon persisted until at least 2 Ga.”*

Line 157. This model age from the literature is different from the model age determined for these basalts in an associated manuscript (Non-KREEP origin for Chang'E-5 basalt in the Procellarum KREEP Terrane”. Why not use the model age measured on these samples? What is the μ value if the older model age is used? What is the interpretation of the model age?

The model age is used to represent the formation time of source reservoirs generated during the crystallisation of the lunar magma ocean (LMO). Though model ages may vary from 4420 Ma to 4300 Ma, the first stage from moon formation to modeled source formation age is much shorter than the second stage (modeled age to 2030 Ma). Thus, selection of different model ages has little impact on the calculated μ . If 4420 Ma is used, the μ is 679, while using 4376 Ma, μ is 684. For direct comparison with previous results, we follow Snape et al. (2016, 2019) and select 4376 ± 18 Ma. Addressing similar comments from both reviewers, we added a paragraph in main text with a detailed description the μ calculation procedure as well as a better explanation of the significance of the μ value for most readers.

What is the origin for the difference in μ values between KREEP and the sources for the mare basalts?

The origin of the difference in μ values between KREEP and the sources for the mare basalts is due to the differentiation and crystallisation of the LMO.

Lines 161-162. Again a subtlety in the interpretation. Heat generated by KREEP may have heated the mantle without being in the sources for these basalts?

Accepted. We rephrased here as: *“.....the results in this study using the radiogenic element U offer direct evidence for heat-producing elements not being concentrated in the Chang'E-5 basalt source. Thus, these results strongly suggest that the idea of KREEP-induced heating for these youngest lunar magmas requires further investigation or the consideration of other mechanisms.”*

Lines 180-182. Not sure what this means. Clearly, extending the crater counting among the Apollo sites provides evidence for the reliability of crater counting on the Moon over a restricted period of time.

This sentence is now rephrased as: *“Therefore, for the first time after the last lunar sample return mission more than 44 years ago, our radioisotope age obtained for the newly-returned Chang'E-5 samples verifies the first-order reliability of the lunar crater-counting chronology established by the Apollo and Luna missions.”*

Charles "Chip" Shearer

We greatly appreciate the constructive comments and suggestions that have greatly improved the manuscript.

Reviewer #2 Evaluations:

This manuscript documents the age and Pb initial isotopic composition of basaltic clasts from lunar soils recently collected by the Chang'E 5 sample return mission. These data provide unprecedented new insight in the chronology of the lunar magmatic activity based on exceptional samples. Furthermore, using their new age, the authors provide further constraints on calibrating the crater counting method.

The manuscript is well written and well organised. The data are solid and the analytical methods used to acquire them as well as data processing protocols are sound. In this latter aspect, the authors did a very good job: they have assessed the terrestrial Pb contamination in the analyses and filtered the dataset accordingly. Failing to do this procedure was a common mistake in the past that crippled many studies based on U-Pb or Pb-Pb dating techniques yet still seen in many recent papers dealing with lunar rocks. This makes this study sound and the ages reliable then worth publishing in Nature.

We appreciate the reviewer's positive evaluation on the scientific value of this work.

Nevertheless, this manuscript is not perfect and needs some improvement.

All the suggestions below have been accepted, and changes are integrated into the revision.

Specifically, I noticed one potential issue concerning the calculation of the Pb initial isotopic composition. This section of the paper is confusing in particular when reading the captions of figure 2. As it stands (as I understand it), the caption and the text suggest that the Pb initial $^{207}\text{Pb}/^{206}\text{Pb}$ ratio was calculated from 6 data. It is not mentioned if the calculation is a weighted average. In this case, the calculated values should be given with relevant statistical parameters and the corresponding plots could be provided in supplementary material.

Yes, it is a weighted average. In the revision, we have made a clear statement, including relevant statistical parameters.

The initial $^{204}\text{Pb}/^{206}\text{Pb}$ ratio was calculated using a 2-stage Pb evolution model and an evolution curve with a μ value of 1000; this evolution curve being the best fit for intersecting the isochron where are located the data with the highest $^{207}\text{Pb}/^{206}\text{Pb}$ ratios (and then, the calculated average initial $^{207}\text{Pb}/^{206}\text{Pb}$ ratio). This μ value must have been estimated before the calculation of the initial $^{204}\text{Pb}/^{206}\text{Pb}$ ratio. But once, this initial $^{204}\text{Pb}/^{206}\text{Pb}$ ratio is calculated, it is used in turn to calculate the μ value of the mantle source, as seen in Source data Table 2. If this is correct, it looks like there is

an issue of circular argument regarding the calculation of the initial $^{204}\text{Pb}/^{206}\text{Pb}$ ratio and μ value of the mantle source. The authors need to either better explain the procedure leading to the calculation of the Pb initial isotopic composition or revise this procedure to avoid this circular argument issue. Perhaps, using a weighted average for the calculation of the $^{204}\text{Pb}/^{206}\text{Pb}$ ratio is the best way despite this could yield a less precise average ratio than less precise μ value. This is probably my main concern about this manuscript.

We are grateful to the reviewer for making this suggestion. We followed the good advice to calculate the $^{204}\text{Pb}/^{206}\text{Pb}$ by a weighted average of measured initial $^{207}\text{Pb}/^{206}\text{Pb}$ with negligible U and the best-fitted isochron. Thus, the two-stage Pb evolution curve is not necessary here. We have deleted it from Figure 2 to make this figure more comprehensible and not cluttered.

I think the manuscript is a bit unbalanced: the discussion about the Pb initial isotopic composition of the Moon is quite small (L161-L168) in comparison to the discussion about calibrating the crater counting method (L175-L199). I think this is related to the fact that the former is a bit underdeveloped while the importance of the latter is a bit overstressed. Calibrating the crater counting method is important but I would say that it is a secondary issue compared to the determination of Pb isotopic composition of lunar basalts then identification of possible mantle reservoirs (including KREEP). The crater counting method is a relative chronology method which is useful as we don't have enough samples to assess the period of activity of lunar magmatism. Its application on other astronomical bodies is in fact limited as we still need a representative set of samples to be calibrated this method (samples we do not have) and of course, a large planetary surface covered by craters. As soon as we will have a large set of samples; this method will most likely be obsolete.

I would encourage the authors to further discuss the implications of having basalts derived from a source with relatively low μ values sampled in Procellarum terrane where basalts are expected to be derived from source with high μ values or try to explain why basalts tend to show minor or none contribution of a KREEP component as they are getting younger.

Agree. In the revision, we modify the calculation of initial Pb isotopic compositions of Chang'E-5 lunar basalts and expand the discussion on the significance of their U-Pb isotopic systematics (μ value) for the chemical origin and evolution of their mantle source.

I also noted few minor issues that listed below:

- Abstract: the mention of the crater counting method and its calibration come on Line 4 while the aspect of Pb isotopic composition of mantle reservoirs is mentioned at the end of the "results" section of the abstract close to the end of it. I would place this aspect of mantle reservoir more at the beginning of the abstract and mention the crater counting calibration at the end of the "results" section. By doing this, the abstract would follow the layout of the manuscript where the discussion about crater counting method is at the end of the paper.

We revised the abstract following Nature's instructions and this comment.

- Source data table 1 needs some improvement. The caption mentions that "shadowed data" were not used for the isochron construction but the data are either in black font or red font so the term "shadowed" seems inadequate. A column with the actual $^{238}\text{U}/^{208}\text{Pb}$ should be included in this table for the plagioclase analyses. There is a column with the ^{208}Pb counts for phosphates: this seems rather weird but is there any reason for that? Finally, some plagioclase data are outlined as used in the calculation of the Pb initial isotopic ratios but among these data; some of them are in black font while others are in red font suggesting that some data rejected for the isochron construction might have been used in the initial ratio calculation. Actually, according to this table, seven plagioclase data seem to have been used in the

calculation of the Pb initial ratio instead of 6 as mentioned in the text. Finally, two data among these plagioclase analyses used to the calculation of the initials (again, if I read correctly the table caption), have U/Pb ratio higher than 0.01. According to the text, these data should have been discarded for the initials calculation. All these little issues should be either fixed.

We appreciate the reviewer's detailed check of our work. We revised the table accordingly. The source data are refined as only data columns used for corresponding Figures. The complete data was provided as supplementary Tables in Excel format. We discarded two data with UO^+/Pb^+ ratio of ~ 0.02 , and calculated the initial $^{207}Pb/^{206}Pb$ with 5 data with $UO^+/Pb^+ < 0.01$. The initial $^{204}Pb/^{206}Pb$ was calculated following the reviewer's good suggestion, i.e., by this initial $^{207}Pb/^{206}Pb$ and the isochron. The new calculation procedure produces almost an identical result to the previous one, but with a little larger uncertainty (μ has changed from 680 ± 20 to 684 ± 42).

- In the supplementary material Tables 1 and 2 (Electron microprobe data), this is a minor detail but when recalculating the totals of the analyses, I found a discrepancy of 0.02- 0.01% between my calculation and the totals indicated in the paper.

This apparent discrepancy was because the original data format was Excel with more significant digits than the data in the Word version of the table. Per Nature's request, supplementary tables must be supplied in Excel format. Thus, there is no such discrepancy in the revision.

- In the supplementary material Table 3, it is not clear why there are two lines for the background noise. Does it mean that these background measurements correspond to two different analytical sessions? In the supplementary material, three analytical sessions are mentioned. This needs to be clarified.

Thanks for your reminder. In the revision, we added the background noise for the third session.

Here below are line-by-line comments on specific issues I spotted across the manuscript:

- L72-L76: I don't really understand this section: at the beginning of the sentence, the authors say they picked clasts from 2 lunar soils but the next part of the same sentence, I understand they mounted another soil sample in epoxy as well as 4 basaltic clasts in in 2 other epoxy mounts. To be mounted in epoxy; the grains in these 3 mounts must have been picked up from bulk soil. This is even more confusing when reaching line 91 where it is said that 47 basaltic clasts were investigated for Pb-Pb dating. The reader needs to know how many different samples scooped from the surface were investigated, how many grains were picked from these samples and how many were mounted in each epoxy mount and if there is a lithology sorting in the different epoxy mounts

The samples we studied were allocated by the China National Space Administration, including two aliquots of soils and three one-inch epoxy mounts. From the soils, we picked hundreds of lithic blasts to make additional epoxy mounts and searched for datable minerals. We rephased this section as follows:

"The Chang'E-5 samples studied in this research were scooped from the lunar surface and include three one-inch epoxy mounts (samples CE5C0000YJYX041GP and CE5C0000YJYX042GP with two basalt clasts larger than 1.5 mm in each, and CE5C0800YJFM00102GP with 20 mg of soil) and two aliquots of soils (samples CE5C0100YJFM00103 of 1 g and CE5C0400YJFM00406 of 2 g) allocated by the China National Space Administration. Around 800 lithic clasts (>0.25 mm) were randomly picked from the two soil samples to make additional epoxy mounts."

- L92: Do the authors have an idea of the bulk chemistry of the clasts: for the EDS maps, a rough estimate of the chemistry is possible. the bottom line is: are the clasts low-Ti, High-Ti, high-Al? or even KREEP?

There is another companion paper with detailed descriptions of the petrology and other geochemistry, including element contents and Sr-Nd isotopes. The Chang'E-5 basalt clasts can be classified as low-Ti/high-Al/low-K type. These basalts have higher FeO (22.2 wt.%), TiO₂ (5.7 wt.%), and Al₂O₃ (11.6 wt.%) contents and lower Mg# (32.1) relative to the Apollo and Luna low-Ti basalts. We focused on the geochronology in this paper.

- L122-123: I am not comfortable with this: the cited reference is a submitted manuscript. I think the authors made their points by saying that the different clasts formed at the same time.

We delete the reference here and argue for the same source based on the data presented in our paper. Accordingly, we revised this as: *“Despite the distinct petrographic textures in Chang'E-5 basalts, these four isochrons have consistent y-intercepts and slopes (Extended Data Fig. 3) within uncertainties, indicating both their identical age and derivation from most likely the same source.”*

- L133-134: The authors could be more explicit here and give a hint about their opinion about what are these new constraints

It is a wise suggestion to expound on the implications here. We rephrased here as: *“This study provides the first conclusive evidence that magmatic activity on the Moon persisted until at least 2 Ga. This new insight into the existence of this youngest known volcanism provides a critical constraint for understanding the thermal mechanisms behind the longevity of lunar magmatism.”*

- L137-138: It looks like this evolution curve has a μ value of 1000. Is that correct? I do not really understand the meaning of the ticks on this curve and of “4376-2030 Ma”: is this the age span of the second stage of Pb evolution?

This blue curve with the ticks followed that of Snape et al. (2016) and illustrated the Pb evolution with different μ (500 and 1000 are labelled) after the second stage with the age span from 4376–2030 Ma. In the revision, as wisely suggested by the reviewer, we calculated the initial ²⁰⁴Pb/²⁰⁶Pb by isochron and ²⁰⁷Pb/²⁰⁶Pb. Thus, we have since deleted this blue curve that had caused confusion.

- L138-140: This sounds like the authors found the best fitting evolution curve which has a μ value of 1000. This best fitting curve being those intersecting the isochron at its extremity (where plot the data with the highest 207/206Pb ratios). Following this, the authors determine the Pb initial isotopic ratios.

As explained above, we have deleted this blue curve and now follow the reviewer's suggestion to calculate the initial ²⁰⁴Pb/²⁰⁶Pb by isochron and ²⁰⁷Pb/²⁰⁶Pb instead.

- L148: "have been calculated" or "can be calculated"?

Accepted. Changed to, *“can be calculated.”*

- L150: It looks like some data in this table used in the calculation of the initial 207/206 ratio have 238U⁺/208Pb⁺ ratio higher than 0.01.

We are thankful for the detailed check. In the revision, we omit two data with UO⁺/208Pb⁺ around 0.02.

- L152-155: This is a weighted average? if so, say it. If the authors do not want to use a weighted average calculation for determination of 204/206 initial ratio, why not using the isochron to determine the 204/206 ratio: the authors have already the 207/206 ratio, it seems straightforward.

Good idea! Following this suggestion, we calculated the initial $^{204}\text{Pb}/^{206}\text{Pb}$ with well-constrained initial $^{207}\text{Pb}/^{206}\text{Pb}$ and the best-fitted isochron (see above).

- L156-158: How this μ value is calculated needs some clarifications: this value cannot be calculated using a 204/206 ratio determined from the intersect between the isochron and an evolution curve with an estimated μ value representing the best fit for such intersect: it is a circular argument issue....

Following this good advice, we calculate the initial $^{204}\text{Pb}/^{206}\text{Pb}$ ratio based on the best-fitted isochron and the measured initial $^{207}\text{Pb}/^{206}\text{Pb}$, not using the intersection between the isochron and the evolution curve. Then, the μ value can be calculated based on a two-stage Pb evolution model suggested by Snape et al. (2016).

- L160-163: This statement is not clear and might be borderline overinterpretation: the data tend to suggest a limited if any involvement of a KREEP component in the geochemical characteristics of the source of the basalts but the authors seem to assume that this is a contribution of a KREEP reservoir in the source of these lunar basalts. This has to be explained or clarified. The authors can outline that no or very little KREEP contribution is quite unusual for basalts from Procellarum terrane which is the locus of KREEP-flavoured basalts. The authors seem to suggest that contamination of Low-Ti basalts en-route to the surface by material with KREEP chemical signature but it is not clear how they reach this conclusion. This has to be explained/demonstrated. Furthermore, the authors might have also to consider how their data fit or do not fit with the hypothesis expressed by Merle et al. (2020) suggesting that the low-Ti basalts seem to be more contaminated by a KREEP component as they are younger.

The reviewer understands correctly that the data indicate a limited, if any, involvement of a KREEP component in the geochemical characteristics of the source of the basalts. Thus, what we had claimed was that the KREEP characteristics of the whole rock were likely acquired later by extensive fractionation during magma crystallization. However, as this is not our focus of this manuscript, we delete this expression in the revision.

Merle et al. (2020) did convincing work to suggest that the low-Ti basalts seem to be the more contaminated by a KREEP component as the younger. We cite this work and make a comparison as follows in main text:

“An apparent increase in μ values from ca. 3.4–3.0 Ga for low-Ti Apollo basalts and low- and ultra-low-Ti basaltic meteorites (NWA 4734 and NWA 773 clan) suggests a progressive contribution of a KREEP-like component in such rocks (Merle et al., 2020). However, the Chang’E-5 basalts do not follow this trend, indicating that KREEP-like components were not involved in our samples. Corroborating evidence for a non-KREEP source for the Chang’E-5 basalts is provided by Sr-Nd isotopes, but the results in this study using the radiogenic element U offer direct evidence for heat-producing elements not being concentrated in the Chang’E-5 basalt mantle source. Thus, these results strongly suggest that the idea of KREEP-induced heating for these youngest lunar magmas requires further investigation or the consideration of other mechanisms.”

- L165-166: The authors could remove this sentence as they cannot rely on the results/interpretations from this manuscript as long as it is not published. It is not very critical anyway.

Accepted. Sentence deleted.

- Figure 3: There are also data of low-Ti basalts from lunar meteorites. There is no obvious reasons to not use these data for comparison.

Accepted. The low-Ti basalts from the lunar meteorites studied by Merle et al. (2020) have been added to Figure 3 for comparison.

- L182: if this is a model age, it cannot be "absolute".

Good catch. "Absolute" deleted.

- L198-199: Please explain. It is obvious for the Moon but for other planets, it is not that clear: except Mars, we don't have much samples from other planets...

We rephrased here. The reviewer is correct that we don't have many samples from other planets. This is precisely why the lunar crater counting chronology is necessary to be improved by returned samples, because the lunar crater counting chronology has been the basis for the translation of the crater counting chronologies of the other terrestrial bodies without returned samples.

- L302: Very minor detail: it is an EDS detector: EDS is energy dispersive x-ray spectroscopy. So it is spectroscopy not spectrometry. Strictly speaking...

We corrected this in the revision.

- L315-316: replace "obtained by" by "processed with".

Accepted.

- L323-324: please give detection limit for major elements.

The detection limit for major elements is now provided in the revision: "...whereas those for major elements (Si, Ti, Al, Fe, Mn, Mg Ca, Nb, P, S, and Cl) are 60-120 ppm."

- L330-332: this sentence could be improved: "the distinct Pb isotopic ratios yielded by these two types of phases help populating the isochron and calculate a precise date".

Accepted.

- L339-341: it looks like this is expressed in the next few lines (L342-L354). The authors could slightly rephrase to make it a bit more straightforward.

Accepted. We rephrased here to make it more straightforward.

- L339: replace "is" by "focused on"

Accepted.

- L349: replace "was" by "were"

Accepted.

- L350: replace "position" by "localise"

Accepted.

- L362-369: Mass resolution and standards used as well as the average values of their isotopic ratios and related analytical precision should be indicated here as for the first analytical session.

Accepted.

- L370-379: standards? Related average value and precision should be also provided.

Accepted.

- L380-381: there are 2 lines with 20 and 30 background measurements in this table. Therefore, it is not clear if these 2 sets correspond to only 2 analytical sessions. This would mean that backgrounds for one analytical session are missing. Indicate clearly the corresponding analytical session for the different sets of background analyses in this table.

We appreciate the reminder. In the revision, we have added the background noise for the third session.

- L389-394: Here the authors should mentioned that the method was first established by Connelly et al., 2012 (Science, 338, P651-655) then applied to SIMS by Snape et al., 2016 and further expressed and refined by Snape et al., 2019 and Merle et al., 2020. Indeed L390-L393 were stated first by Connelly et al., 2012.

Accepted. We revised here as suggested.

- L407-410: Here cite Snape et al., 2019 and Merle et al., 2020/ these 2 papers clearly expressed this procedure.

Accepted.

- L410-411: Did the authors calculate a weighted average for these values?

Not yet. The U contents of different grains show a wide range based on the Pb intensities. A weighted average may not reflect this wide variation.

As a conclusion, there are only few issues that are relatively easy to fix but this requires an extra bit of work before publication.

We greatly appreciate the constructive comments and suggestions that have greatly improved the manuscript.

Best wishes, Renaud Merle

Reviewer Reports on the First Revision:

Referee #1:

The author's responded to my original review in a very positive manner. I don't think additional minor comments would be helpful to producing further improvements to this manuscript. Again, I strongly suggest that this manuscript be published in its current form.

Referee #2

In this revised version of their manuscript, the authors have properly answered to all the comments I made and fixed all the issues I outlined. The manuscript is now flawless and ready for publication. I made few more suggestions which are not critical but I believe would help improving the text. I listed them line by line below.

In the abstract (L27), I would not say "two-stages Mu value" but rather "a mu value of 680 for a source that evolved through 2 stages of differentiation".

L50: I think is needed "in" after "accumulated".

L61-67: still a bit confusing and complex. I would suggest: "5 epoxy mounts were made: 2 with basalt clasts and 3 with soil". It is not clear what the authors mean by "soil aliquots". What is the difference between the soil sample CE5C0800YJFM00102GP and the 2 other soil aliquots (CE5C0100YJFM00103 and CE5C0400YJFM00406)?

L67: perhaps say: "the lithic clasts in the soils".

L91-92: I prefer expressing old ages in ma rather than in Ga as the uncertainty on the age when very small like in this paper, is more highlighted. On L92, replace Ma by Ga (typo).

L95: replace Ma by Ga.

L99: What do the authors mean by "integrated" isochron?

L100: typo: Ma for Ga

L117: remove "essential" and say "rock-forming minerals" or "major phase".

L123-L126: I would suggest the authors use first the weighted mean for calculating the 204/206 ratio and using the lowest measured 204/206 ratios (as the authors did) then confirm this calculation using the graphical determination of the 204/206 ratio. The weighted mean for 204/206 is actually quite precise and its uncertainty is not different from other initials for

Apollo or meteorite samples. Most of the calculated 204/206 ratios from Apollo and meteorites were calculated that way even if individual analyses were not very precise. So it makes possible a direct comparison between these data without any problem. I would also insert in the supporting material, the weighted average plots.

L130-131: I would suggest "Determination of the μ value of the mantle source is dependant of the Pb evolution model".

L145: The authors should emphasize the fact that the contribution of KREEP component in the VLT and low-Ti is stronger as the basalts are younger. If stated as previously, the authors can highlight efficiently that the Chang'E 5 samples are very unique as they do not follow the trend of the younger VLT and low-Ti basalts.

L156-158: This sentence does not present efficiently the arguments in favour of a mare origin for the Chang'E 5 samples. The authors should be more explicit.

L158-161: I would say that the samples offer the unique opportunity to confirm the crater-counting chronology on the Moon.

Fig 2: Perhaps add an insert in fig 2a to show that it represents fig 2b. Usually, I prefer to plot error bars at 2 σ level. In particular as the authors plotted data in fig. 3 at 2 σ level. The authors should add the probability of fit P with MSWD.

Fig 2B: typo Ga instead of Ma.

Fig. 4: Error bars are plotted as 1 σ level again. The authors should be consistent and plot everything with the same level of uncertainty.

Supplementary material table S3: I noticed 3 plag data that were shadowed then rejected for the isochron construction but mentioned as used for the calculation of the Pb initial ratios. Here below, I listed these data.

```
CE5C0100YJFM00103-001-PI@5  
CE5C0100YJFM00103-001-PI@8  
CE5C0400YJFM00406-002-PI@3
```

This leads to a serious concern: Did the authors rejected these data from the isochron construction but used them anyway for the calculation of the Pb initial ratios? The point is: among the data used for the isochron construction, those from Plag or K-feldspars with the highest 207/206Pb ratios are expected to represent the Pb initial composition of the sample. When several analyses of these phases yielded ratios similar within uncertainties, a weighted average can be calculated.

Data rejected from the isochron cannot be used for the determination of initial ratios as they are suspected to be contaminated by terrestrial Pb then the ratios, biased.
The authors have to check if this is a typo or a more serious issue.

Once these little issues are fixed, the paper will be ready for publication.

Best regards,

Renaud Merle

Author Rebuttals to First Revision:

Reviewer #1 Evaluations:

The author's responded to my original review in a very positive manner. I don't think additional minor comments would be helpful to producing further improvements to this manuscript. Again, I strongly suggest that this manuscript be published in its current form.

We appreciate the reviewer's positive evaluation of the scientific value of this work.

Reviewer #2 Evaluations:

In this revised version of their manuscript, the authors have properly answered to all the comments I made and fixed all the issues I outlined. The manuscript is now flawless and ready for publication. I made few more suggestions which are not critical but I believe would help improving the text. I listed them line by line below.

We appreciate the reviewer's positive evaluation of the scientific value of this work.

In the abstract (L27), I would not say "two-stages Mu value" but rather "a mu value of 680 for a source that evolved through 2 stages of differentiation".

Accepted. We revised here as suggested.

L50: I think is needed "in" after "accumulated".

We rephased here to "which have incurred fewer impacts."

L61-67: still a bit confusing and complex. I would suggest: "5 epoxy mounts were made: 2 with basalt

clasts and 3 with soil". It is not clear what the authors mean by "soil aliquots". What is the difference between the soil sample CE5C0800YJFM00102GP and the 2 other soil aliquots (CE5C0100YJFM00103 and CE5C0400YJFM00406)?

The China National Space Administration mixed the scooped soil thoroughly then subpackaged into ten bottles equally, labelled as CE5C01 ~ CE5C10. The studied soils are from the CE5C01, CE5C04, CE5C08 bottles, respectively. Thus, there should be no difference between different aliquots.

L67: perhaps say: "the lithic clasts in the soils".

Accepted. We revised here as suggested.

L91-92: I prefer expressing old ages in ma rather than in Ga as the uncertainty on the age when very small like in this paper, is more highlighted. On L92, replace Ma by Ga (typo).

We agree. In fact, we already used Ma. Thousands are separated by commas and you may have mistook the comma as a dot. Here, "Ma" is right.

L95: replace Ma by Ga.

Thousands are separated by commas and you may have mistook the comma as a dot. Here, "Ma" is right.

L99: What do the authors mean by "integrated" isochron?

Firstly, we constructed 4 isochrons individually for each type of basalt texture. Then, considering their consistent ages and isochrons slopes, the four isochrons were integrated as one isochron. This is explained in the text.

L100: typo: Ma for Ga

Thousands are separated by commas and you may have mistook the comma as a dot. Here, "Ma" is right.

L117: remove "essential" and say "rock-forming minerals" or "major phase".

Accepted. We revised here as "rock-forming minerals".

L123-L126: I would suggest the authors use first the weighted mean for calculating the 204/206 ratio and using the lowest measured 204/206 ratios (as the authors did) then confirm this calculation using

the graphical determination of the $^{204}\text{Pb}/^{206}\text{Pb}$ ratio. The weighted mean for $^{204}\text{Pb}/^{206}\text{Pb}$ is actually quite precise and its uncertainty is not different from other initials for Apollo or meteorite samples. Most of the calculated $^{204}\text{Pb}/^{206}\text{Pb}$ ratios from Apollo and meteorites were calculated that way even if individual analyses were not very precise. So it makes possible a direct comparison between these data without any problem. I would also insert in the supporting material, the weighted average plots.

You are right that there are two spots with a consistent measured $^{204}\text{Pb}/^{206}\text{Pb}$ ratio with the constrained value based on the best-fit Pb-Pb isochron and the best estimate of the initial $^{207}\text{Pb}/^{206}\text{Pb}$ ratio. The calculated $^{204}\text{Pb}/^{206}\text{Pb}$ value is therefore more precise and adopted here.

L130-131: I would suggest “Determination of the μ value of the mantle source is dependant of the Pb evolution model”.

Accepted. We revised here as suggested.

L145: The authors should emphasize the fact that the contribution of KREEP component in the VLT and low-Ti is stronger as the basalts are younger. If stated as previously, the authors can highlight efficiently that the Chang'E 5 samples are very unique as they do not follow the trend of the younger VLT and low-Ti basalts.

You are right that the Chang'E-5 samples do not follow the trend. However, it may not be unique, as some lunar basalt meteorites also do not follow the trend.

L156-158: This sentence does not present efficiently the arguments in favour of a mare origin for the Chang'E 5 samples. The authors should be more explicit.

Accepted. We now extend the introduction about the homogeneity of the sampled mare unit as seen from orbit.

L158-161: I would say that the samples offer the unique opportunity to confirm the cratercounting chronology on the Moon.

Accepted. “unique opportunity” is added.

Fig 2: Perhaps add an insert in fig 2a to show that it represents fig 2b. Usually, I prefer to plot error bars at 2S level. In particular as the authors plotted data in fig. 3 at 2S level. The authors should add the probability of fit P with MSWD.

We described Fig.2b in legends as the enlarged lowest part of the isochron in Fig.2a. So, it is not nesscessary to add an insert in Fig.2 a.

Fig 2B: typo Ga instead of Ma.

Thousands are separated by commas and you may have mistook the comma as a dot. Here, “Ma” is right.

Fig. 4: Error bars are plotted as 1S level again. The authors should be consistent and plot everything with the same level of uncertainty.

The values are cited from references at the confidence level at which they were originally reported. In order to compare with other references, we keep this level here, and provide a clear statement in the legends.

Supplementary material table S3: I noticed 3 plag data that were shadowed then rejected for the isochron construction but mentioned as used for the calculation of the Pb initial ratios. Here below, I listed these data.

CE5C0100YJFM00103-001-PI@5

CE5C0100YJFM00103-001-PI@8

CE5C0400YJFM00406-002-PI@3

This leads to a serious concern: Did the authors rejected these data from the isochron construction but used them anyway for the calculation of the Pb initial ratios? The point is: among the data used for the isochron construction, those from Plag or K-feldspars with the highest $^{207}\text{Pb}/^{206}\text{Pb}$ ratios are expected to represent the Pb initial composition of the sample. When several analyses of these phases yielded ratios similar within uncertainties, a weighted average can be calculated. Data rejected from the isochron cannot be used for the determination of initial ratios as they are suspected to be contaminated by terrestrial Pb then the ratios, biased. The authors have to check if this is a typo or a more serious issue.

We appreciate the reviewer's carefulness. The rationale behind the reviewer's concern is reasonable, however, its impact is negligible here for the Change'E-5 basalts. The initial $^{206}\text{Pb}/^{204}\text{Pb}$ is around 430 and the $^{207}\text{Pb}/^{206}\text{Pb}$ is ~ 0.86 , while for Pb from terrestrial contamination, the $^{206}\text{Pb}/^{204}\text{Pb}$ is around 18 and $^{207}\text{Pb}/^{206}\text{Pb}$ is ~ 0.83 . Therefore, in the event that contaminated ^{204}Pb was doubled or tripled, the apparent $^{204}\text{Pb}/^{206}\text{Pb}$ would be also doubled or tripled. But the $^{207}\text{Pb}/^{206}\text{Pb}$ ratios are only biased less than 0.3%, much below the measured error of $>3\%$ level. So, these three analyses with lowest U/Pb (Extended Data Fig.4) can provide a valuable estimate for initial $^{207}\text{Pb}/^{206}\text{Pb}$.

Once these little issues are fixed, the paper will be ready for publication.

Best regards,

Renaud Merle